# BaCaDI: Bayesian Causal Discovery with Unknown Interventions

**Alexander Hägele**[1]     **Jonas Rothfuss**[1]     **Lars Lorch**[1]     **Vignesh Ram Somnath**[1,3]     **Bernhard Schölkopf**[1,2]

**Andreas Krause**[1]

[1]Department of Computer Science, ETH Zürich, Switzerland
[2]Max Planck Institute for Intelligent Systems, Tübingen, Germany,
[3]IBM Research Zürich, Switzerland

## Abstract

Learning causal structures from observation and experimentation is a central task in many domains. For example, in biology, recent advances allow us to obtain single-cell expression data under *multiple interventions* such as drugs or gene knockouts. However, a key challenge is that often the *targets of the interventions are uncertain or unknown*. Thus, standard causal discovery methods can no longer be used. To fill this gap, we propose a Bayesian framework (BaCaDI) for discovering the causal structure that underlies data generated under various unknown experimental/interventional conditions. BaCaDI is fully *differentiable* and operates in the continuous space of latent probabilistic representations of both causal structures and interventions. This enables us to approximate complex posteriors via gradient-based variational inference and to reason about the epistemic uncertainty in the predicted structure. In experiments on synthetic causal discovery tasks and simulated gene-expression data, BaCaDI outperforms related methods in identifying causal structures and intervention targets. Finally, we demonstrate that, thanks to its rigorous Bayesian approach, our method provides well-calibrated uncertainty estimates.

## 1 INTRODUCTION

Identifying causal dependencies by empirical observation and experimentation is a problem of fundamental scientific interest. If we understand the causal mechanisms that govern a system of interest, we can predict its behavior when parts of system are *actively manipulated* from outside. For instance, if we understand the causal structure of a gene regulatory network, we can predict the effect of drugs or gene knockouts more reliably [Meinshausen et al., 2016]. While identifying the true causal structure from observational data

is impossible in many cases [Pearl, 2009, Peters et al., 2017], intervening on some variables in the system and observing the outcome can provide valuable information helping with identification.

In some settings, we can collect a range of datasets from the same causal system under multiple different interventions. For instance, recent advances in biology allow us to obtain single-cell gene expression data under interventions such as different drug candidates or gene knockouts [Srivatsan et al., 2020, McFarland et al., 2020]. Causal discovery methods that can work with interventional data typically assume knowledge of the targets, i.e., which variables have been intervened upon, or the statistical effect of the interventions [Hauser and Bühlmann, 2012, Wang et al., 2017, Yang et al., 2018]. However, this limits their applicability because the intervention targets and effect are often *uncertain or unknown* in practice, for instance, due to unknown off-target effects upon drug administration. An additional challenge is that we typically only have a *small number of data points per intervention* due to high experimental costs. This renders joint inference over causal structures and interventions brittle, and a rigorous treatment of uncertainty is paramount for reaching reliable conclusions. Recent methods [Mooij et al., 2016, Ke et al., 2019, Brouillard et al., 2020, Squires et al., 2020] that are able to deal with imperfect or unknown interventions do not account for epistemic uncertainty during inference. Thus, they frequently fail in realistic scenarios where interventional data is scarce.

Addressing these shortcomings, we introduce *Bayesian Causal Discovery with unknown Interventions (BaCaDI)*, a fully Bayesian approach for inferring the complete set of causal structure, functional mechanisms and intervention targets given various unknown experimental/interventional conditions. In particular, BaCaDI performs joint inference over causal Bayesian Networks (CBNs) as well as intervention targets and effects across multiple experimental contexts. Our method is fully *differentiable* and operates in the continuous space of latent probabilistic representations of both BNs and interventions. This makes our approach

*Accepted for the Causal Representation Learning workshop at the 38th Conference on Uncertainty in Artificial Intelligence* (UAI CRL 2022).

particularly scalable to causal systems of many variables, and it enables us to approximate complex posteriors via gradient-based particle variational inference.

In a range of experiments with synthetic causal BNs as well as a realistic gene-expression simulator, BaCaDI outperforms related methods in recovering the underlying causal structure and intervention targets. Moreover, the principled treatment of uncertainty allows BaCaDI to provide well-calibrated uncertainty estimates for its predictions, even when the underlying Bayesian model is misspecified as in the case of gene-expression data.

## 2 RELATED WORK

**Causal discovery from multiple contexts.** Learning a causal structure from datasets collected from different interventional contexts of the same causal system has been referred to as Joint Causal Inference (JCI) [Mooij et al., 2016]. Multiple methods handle such combinations of observational and interventional data [Magliacane et al., 2016, Zhang et al., 2017, Hauser and Bühlmann, 2012, Wang et al., 2017, Yang et al., 2018] but assume full knowledge of the interventions. Other methods build on the notion of *invariance* [Schölkopf et al., 2012, Peters et al., 2016, Meinshausen et al., 2016, Ghassami et al., 2017, Heinze-Deml et al., 2018, Huang et al., 2020], but either do not generalize to full graphs or make restrictive assumptions about the local causal effects.

**Causal discovery with unknown interventions.** Mooij et al. [2016] and Squires et al. [2020] reduce the unknown intervention setting to standard causal discovery tools such as conditional independence and invariance tests. However, such hypothesis tests typically require large datasets, making these methods brittle for realistic datasets of small size. Another line of recent work formulates continuous relaxations of the joint inference problem under unknown interventions and uses gradient-based optimization to find the graph and intervention targets [Ke et al., 2019, Brouillard et al., 2020, Faria et al., 2022]. Our approach uses similar ideas to relax the semi-discrete inference problem into a continuous one. However, we adopt a fully Bayesian approach that enables principled uncertainty quantification and does not only give point estimates. Eaton and Murphy [2007] is the only work we are aware of that does the same, though only for discrete variables and with a costly dynamic programming approach that does not scale well to larger graphs.

## 3 BACKGROUND: CAUSAL DISCOVERY

**Causal Bayesian Networks.** A Bayesian network (BN) $(\mathbf{G}, \mathbf{\Theta})$ uses a directed acyclic graph (DAG) to model the joint density $p(\mathbf{x})$ of $d$ variables $\mathbf{x} = x_{1:d}$ via conditional probabilities. The joint distribution $p$ factorizes as $p(x_1, \ldots, x_d | \mathbf{\Theta}, \mathbf{G}) = \prod_{i=1}^{d} p_i(x_i | x_{\mathrm{pa}_{\mathbf{G}}(i)}, \mathbf{\Theta})$ where

$\mathrm{pa}_{\mathbf{G}}(i)$ is the set of parents of node $i$ in $\mathbf{G}$ and the parameters $\mathbf{\Theta}$ describe the exact local conditional distributions. In a *causal* BN (CBN), the edges also describe direct causal relations. For causal structure learning, we assume that there are no hidden confounding variables (causal sufficiency) [Pearl, 2009, Spirtes et al., 2000, Peters et al., 2017].

**Interventions.** In the causal graph, an intervention on the variable $i$ corresponds to changing the conditional distribution $p_i$ and replacing it by a new distribution $p_i^I$. An intervention is typically considered imperfect (soft) if the distribution is changed but the dependence on the causal parents remains, or perfect (hard, structural) if all dependencies to the causal parents are removed, resulting in a *mutilated graph* $\mathbf{G}^{I_k}$ [e.g. Pearl, 2009, Peters et al., 2017].

In this work, we assume the setting of a collection of $M$ interventions $\mathcal{I} := (I_1, \ldots I_M)$, where each intervention $I_k := (I_k^{\mathrm{tar}}, \mathbf{\Theta}_{I_k})$ acts on a set of targets $I_k^{\mathrm{tar}} \subseteq \{1, \ldots, d\}$. We use $\mathbf{\Theta}_{I_k}$ to denote parameters that describe the conditional distributions $p_i^{I_k}(x_i | x_{\mathrm{pa}_{\mathbf{G}}(i)}, \mathbf{\Theta}_{I_k})$ induced by the intervention $I_k$ on the target variables $\{x_i \mid i \in I_k^{\mathrm{tar}}\}$. To keep the notation and following exposition simple, we assume perfect interventions, i.e., $p_i^{I_k}(x_i | x_{\mathrm{pa}_{\mathbf{G}}(i)}, \mathbf{\Theta}_{I_k}) = p_i^{I_k}(x_i | \mathbf{\Theta}_{I_k})$. However, all the arguments made in the remainder of the paper also hold for soft interventions. The full data distribution under intervention $I_k$ factorizes into the observational and interventional conditionals:

$$p(\mathbf{x} | \mathbf{\Theta}, \mathbf{G}, I_k) = \prod_{i \notin I_k} p_i(x_i | x_{\mathrm{pa}_{\mathbf{G}}(i)}, \mathbf{\Theta}) \prod_{i \in I_k} p_i^{I_k}(x_i | \mathbf{\Theta}_{I_k})$$

The local conditional distributions of the variables that are *not* intervened upon do not change with respect to the observational distribution, a principle often referred to as invariance [Peters et al., 2016] or modularity [Peters et al., 2017].

## 4 BAYESIAN CAUSAL DISCOVERY WITH MULTI-CONTEXT DATA

**Problem Statement.** In this section, we develop a method for Bayesian inference of the causal Bayesian Network (CBN) given *multiple interventional datasets* generated from the same underlying causal model. This setting is also known as learning from multiple contexts Mooij et al. [2016]. Formally, we are given a set of $M$ datasets $\mathbf{D} = \{\mathcal{D}_1, \ldots, \mathcal{D}_M\}$ with corresponding (unknown) interventions $\mathcal{I} = \{I_1, \ldots, I_M\}$. Each $\mathcal{D}_k$ is a set of independent samples $\mathcal{D}_k = \{\mathbf{x}^{(k,1)}, \ldots, \mathbf{x}^{(k,n_k)}\}$ obtained from the *interventional* data distribution $p(\mathbf{x} | \mathbf{\Theta}_{\mathrm{gt}}, \mathbf{G}_{\mathrm{gt}}, I_k)$ based on the ground truth CBN $(\mathbf{G}_{\mathrm{gt}}, \mathbf{\Theta}_{\mathrm{gt}})$. Observational data, if available, can be added to $\mathbf{D}$ as $\mathcal{D}_0 := \mathcal{D}$ with intervention targets $I_0 = \emptyset$.

Our goal is to infer the ground truth CBN $(\mathbf{G}_{\mathrm{gt}}, \mathbf{\Theta}_{\mathrm{gt}})$ and interventions $\mathcal{I}$ given the dataset $\mathbf{D}$. Compared to standard causal inference, the key difficulty is that in addition to the

ground truth CBN, the intervention targets (i.e $I_k^{\text{tar}}$) and their statistical effects (i.e. $\boldsymbol{\Theta}_{I_k}$) are unknown. Therefore, we need to perform joint inference over $M$ mutilated graphs that are all closely connected to an unknown "prototype" graph. Naturally, such inference is well-posed only if the structural changes implied by each intervention $I_k$ are sparse compared to the overall size of $\mathbf{G}_{\text{gt}}$, i.e. $|I_k| \ll d$.

In many relevant application domains, such as biology, the observed samples $n_k$ are noisy and small in number. Hence, it is paramount to not only to predict a single prototype CBN alongside one intervention hypothesis per dataset, but to also to reason about the *epistemic uncertainty* of our empirical inferences. Such uncertainty estimates allow us to quantify the reliability of our predictions and can be used to actively design future experiments. We thus approach the problem from a Bayesian perspective. This renders the task of learning from $\mathbf{D}$ as a posterior inference problem, which we are going to gradually build up in the following.

**Known interventions.** When the intervention targets $I_k^{\text{tar}}$ and the parameters $\boldsymbol{\Theta}_{I_k}$ of the intervention effect are known, for $k = 1, \ldots, M$, the posterior over CBNs includes (i.) the product of data likelihoods over all datasets in $\mathbf{D}$, and (ii.) interventional instead of observation likelihoods for $\mathcal{D}_{k \geq 1}$,

$$
p(\mathbf{G}, \boldsymbol{\Theta}|\mathbf{D}, \mathcal{I}) \propto \underbrace{p(\mathbf{G})p(\boldsymbol{\Theta}|\mathbf{G})}_{\text{priors}} \underbrace{p(\mathcal{D}_0|\boldsymbol{\Theta}, \mathbf{G})}_{\text{obs. likelihood}}
$$
$$
\cdot \prod_{k=1}^{M} \underbrace{p(\mathcal{D}_k|\boldsymbol{\Theta}, \mathbf{G}, I_k)}_{\text{interv. likelihood}}, \tag{1}
$$

where $p(\mathcal{D}_k|\boldsymbol{\Theta}, \mathbf{G}, I_k) = \prod_{i=1}^{n_k} p(\mathbf{x}^{(k,i)}|\boldsymbol{\Theta}, \mathbf{G}, I_k)$ is the interventional likelihood for $\mathcal{D}_k$ given $I_k = (I_k^{\text{tar}}, \boldsymbol{\Theta}_{I_k})$.

**Unknown interventions.** We include unknown interventions in our inference model by introducing additional priors $p(I_k^{\text{tar}})$ and $p(\boldsymbol{\Theta}_{I_k}|I_k^{\text{tar}})$. Accordingly, the modified posterior follows as

$$
p(\mathbf{G}, \boldsymbol{\Theta}, \mathcal{I}|\mathbf{D}) \propto \underbrace{p(\mathbf{G})p(\boldsymbol{\Theta}|\mathbf{G})}_{\text{priors}} \underbrace{p(\mathcal{D}_0|\boldsymbol{\Theta}, \mathbf{G})}_{\text{obs. likelihood}}
$$
$$
\cdot \prod_{k=1}^{M} \underbrace{p(I_k^{\text{tar}})p(\boldsymbol{\Theta}_{I_k}|I_k^{\text{tar}})}_{\text{interv. priors}} \underbrace{p(\mathcal{D}_k|\boldsymbol{\Theta}, \mathbf{G}, I_k)}_{\text{interv. likelihood}} \tag{2}
$$

The prior $p(I_k^{\text{tar}})$ over intervention targets can incorporate prior beliefs about the structure of interventions, e.g. that only a *sparse* set of variables are subject to an intervention at the same time. The parametrization of the interventional distributions $p_i^I(x_i|\boldsymbol{\Theta}_I)$ is informed by the application, and reflects the general nature of interventions, e.g., gene knockdowns in biology.

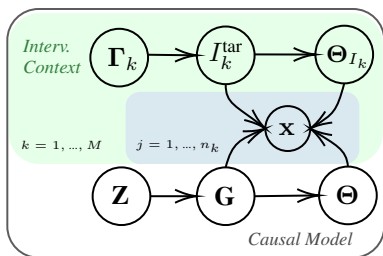

Figure 1: Generative model of combining causal BNs with interventions from different contexts.

# 5 A DIFFERENTIABLE GENERATIVE MODEL OVER CBNS AND INTERVENTIONS

In the following, we represent $\mathbf{G} \in \{0,1\}^{d \times d}$ as the adjacency matrix and $I_k^{\text{tar}} = [I_{k,1}^{\text{tar}}, \ldots, I_{k,d}^{\text{tar}}]^{\top} \in \{0,1\}^d$ as the indicator vector where $I_{k,l}^{\text{tar}} = 1$ if the $l$-th variable is intervened upon and $I_{k,l}^{\text{tar}} = 0$ otherwise. We write $\mathcal{I}^{\text{tar}}$ short for the stack $[I_1^{\text{tar}}, \ldots, I_M^{\text{tar}}]$ of intervention target masks and $\boldsymbol{\Theta}_{\mathcal{I}} := [\boldsymbol{\Theta}_{I_1}, \ldots, \boldsymbol{\Theta}_{I_M}]$ similarly for the intervention effect parameters. The posterior in Eq. 2 is highly intractable as the number of possible DAGs $\mathbf{G}$ grows super-exponentially with the number of variables $d$ [Robinson, 1973]. Furthermore, the number of possible intervention targets $I_k^{\text{tar}}$ grows in the order of $\mathcal{O}(2^d)$.

To facilitate approximate inference using Eq. 2, we harness recent advances in Bayesian structure learning proposed by Lorch et al. [2021] that allows for a *fully differentiable* posterior over graphs. By computing the score of the posterior, we can use approximate inference methods such as variational inference [Blei et al., 2017] or Stein Variational Gradient Descent (SVGD) [Liu and Wang, 2016].

The key idea is to transform the inference problem over discrete structures $\mathbf{G}$ and $I_k^{\text{tar}}$ into one over continuous parameters with respect to which we can differentiate. To this end, we introduce continuous latent variables $\mathbf{Z}$ and $\boldsymbol{\Gamma}_k$ for $k = 1, \ldots, M$ that model the generative processes of $\mathbf{G}$ and $I_k^{\text{tar}}$ through $p(\mathbf{G}|\mathbf{Z})$ and $p(I_k^{\text{tar}}|\boldsymbol{\Gamma}_k)$. This implies the following factorization of the generative model in Figure 1:

$$
p(\mathbf{Z}, \mathbf{G}, \boldsymbol{\Theta}, \boldsymbol{\Gamma}, \mathcal{I}, \mathbf{D}) = \underbrace{p(\mathbf{Z})p(\mathbf{G}|\mathbf{Z})p(\boldsymbol{\Theta}|\mathbf{G})}_{\text{gen. process CBN}} \tag{3}
$$
$$
\cdot \prod_{k=1}^{M} \underbrace{p(\boldsymbol{\Gamma}_k)p(I_k^{\text{tar}}|\boldsymbol{\Gamma}_k)p(\boldsymbol{\Theta}_{I_k}|I_k^{\text{tar}})}_{\text{gen. process interv.}} \underbrace{p(\mathcal{D}_k|\mathbf{G}, \boldsymbol{\Theta}, I_k^{\text{tar}}, \boldsymbol{\Theta}_{I_k})}_{\text{interv. likelihood}}
$$

where we write $\boldsymbol{\Gamma} := [\boldsymbol{\Gamma}_1, \ldots, \boldsymbol{\Gamma}_M]$ for brevity. Under this generative model, it holds that

$$
\mathbb{E}_{p(\mathbf{G}, \boldsymbol{\Theta}, \mathcal{I}|\mathbf{D})}[f(\mathbf{G}, \boldsymbol{\Theta}, \mathcal{I})] = \tag{4}
$$
$$
\mathbb{E}_{p(\mathbf{Z}, \boldsymbol{\Theta}, \boldsymbol{\Gamma}, \boldsymbol{\Theta}_{\mathcal{I}}|\mathbf{D})} \left[ \frac{\mathbb{E}_{p(\mathbf{G}|\mathbf{Z}), p(\mathcal{I}^{\text{tar}}|\boldsymbol{\Gamma})}[f(\mathbf{G}, \boldsymbol{\Theta}, \mathcal{I}) \cdot \mathbf{H}]}{\mathbb{E}_{p(\mathbf{G}|\mathbf{Z}), p(\mathcal{I}^{\text{tar}}|\boldsymbol{\Gamma})}[\mathbf{H}]} \right]
$$

where $\mathbf{H} = p(\boldsymbol{\Theta}|\mathbf{G})p(\boldsymbol{\Theta}_{\mathcal{I}}|\mathcal{I}^{\text{tar}})p(\mathbf{D}|\mathbf{G}, \mathcal{I}, \boldsymbol{\Theta})$ with $p(\mathbf{D}|\mathbf{G}, \mathcal{I}, \boldsymbol{\Theta}) = \prod_{k=1}^{M} p(\mathcal{D}_k|\mathbf{G}, \boldsymbol{\Theta}, I_k^{\text{tar}}, \boldsymbol{\Theta}_{I_k})$ and

$p(\mathcal{I}|\boldsymbol{\Gamma}) = \prod_{k=1}^M p(I_k^{\text{tar}}|\boldsymbol{\Gamma}_k)p(\boldsymbol{\Theta}_{I_k}|I_k^{\text{tar}})$. A proof is provided in Appx. A.1. Eq. 4 allows us to compute Bayesian expectations under the posterior in Eq. 1 in terms of the posterior $p(\mathbf{Z}, \boldsymbol{\Theta}, \boldsymbol{\Gamma}, \boldsymbol{\Theta}_{\mathcal{I}}|\mathbf{D})$ over parameters that are continuous. Before we discuss how to perform approximate inference with $p(\mathbf{Z}, \boldsymbol{\Theta}, \boldsymbol{\Gamma}, \boldsymbol{\Theta}_{\mathcal{I}}|\mathbf{D})$, we further specify some of the conditional probabilities of our generative model and how to make them differentiable.

**Generative model of DAGs G.** Following Lorch et al. [2021], we define the latent variable $\mathbf{Z} = [\mathbf{U}, \mathbf{V}]$ as a stack of two embedding matrices $\mathbf{U}, \mathbf{V} \in \mathbb{R}^{d \times d}$ and the generative model for the adjacency matrix $\mathbf{G}$ by using the inner product between the latent variables in $\mathbf{Z}$:

$$p_\alpha(\mathbf{G}|\mathbf{Z}) = \prod_{i=1}^d \prod_{j \neq i}^d p_\alpha(g_{ij}|\mathbf{u}_i, \mathbf{v}_j) \tag{5}$$
$$\text{with} \quad p_\alpha(g_{ij} = 1|\mathbf{u}_i, \mathbf{v}_j) = \sigma_\alpha(\mathbf{u}_i^\top \mathbf{v}_j)$$

where $\sigma_\alpha(x) = 1/(1 + \exp(-\alpha x))$ is the sigmoid function with inverse temperature $\alpha$ and $\mathbf{u}_i, \mathbf{v}_j$ the $i$-th and $j$-th column vector of $\mathbf{U}$ and $\mathbf{V}$ respectively. We denote the matrix of edge probabilities in $\mathbf{G}$ given $\mathbf{Z}$ by $\mathbf{G}_\alpha(\mathbf{Z}) \in [0,1]^{d \times d}$ with $\mathbf{G}_\alpha(\mathbf{Z})_{ij} := \sigma_\alpha(\mathbf{u}_i^\top \mathbf{v}_j)$. The prior over $\mathbf{Z}$ utilizes (i.) factorized Gaussians with a variance of $\eta_Z^2 = 1/d$ to ensure well-behaved gradients and (ii.) an acyclicity prior using that penalizes the expected cyclicity of $\mathbf{G}$ given $\mathbf{Z}$:

$$p_\beta(\mathbf{Z}) = p(\mathbf{U}, \mathbf{V}) \propto \underbrace{\exp\left(-\beta \mathbb{E}_{p(\mathbf{G}|\mathbf{Z})}[h(\mathbf{G})]\right)}_{\text{acyclicity prior}}$$
$$\cdot \prod_{i=1}^d \underbrace{\mathcal{N}(\mathbf{u}_i|\mathbf{0}, \eta_Z^2 \mathbf{I})\mathcal{N}(\mathbf{v}_i|\mathbf{0}, \eta_Z^2 \mathbf{I})}_{\text{numerical stability}} \tag{6}$$

Here, $\beta$ is the inverse temperature parameter controlling how strongly the acyclicity is enforced, and $h(\mathbf{G}) = \text{tr}\left[(I + \frac{1}{d}\mathbf{G})^d\right] - d \geq 0$. We make use of Theorem 1 in Yu et al. [2019] states that $\mathbf{G}$ is acyclic iff $h(G) = 0$. As $\beta \to \infty$, the support of $p(\mathbf{Z})$ reduces to all $\mathbf{Z}$ that model DAGs [cf. Lorch et al., 2021].

**Generative model of intervention targets $\mathcal{I}^{\text{tar}}$.** Similar to the generative model of $\mathbf{G}$, we define the latent variables $\boldsymbol{\Gamma} \in \mathbb{R}^d$ as the logits of independent Bernoulli distributions that model the entries of the intervention target mask $\mathcal{I}^{\text{tar}} = [I_1^{\text{tar}}, ..., I_M^{\text{tar}}] \in \{0,1\}^{M \times d}$:

$$p(\mathcal{I}^{\text{tar}}|\boldsymbol{\Gamma}) = \prod_{k=1}^M \prod_{i=1}^d p_\alpha(I_{k,i}^{\text{tar}}|\gamma_{k,i}) \tag{7}$$
$$\text{with} \quad p_\alpha(I_{k,i}^{\text{tar}} = 1|\gamma_{k,i}) = \sigma_\alpha(\gamma_{k,i})$$

We denote the matrix of intervention target probabilities as $\mathcal{I}_\alpha^{\text{tar}}(\boldsymbol{\Gamma}) \in [0,1]^{M \times d}$ with $\mathcal{I}_\alpha^{\text{tar}}(\boldsymbol{\Gamma})_{k,i} = \sigma_\alpha(\gamma_{k,i})$. Similar to $\mathbf{Z}$, the prior over $\boldsymbol{\Gamma}$ has three components: (i.) A Gaussian term for *numerical stability*, (ii.) a Beta-distribution prior that encourages $\sigma_\alpha(\gamma_{k,i})$ to be close to 0 or 1, and (iii.) a *sparsity prior* with the $l_1$-norm of $\sigma_\alpha(\boldsymbol{\Gamma}_k)$ and the inverse temperature parameter $\lambda$.

$$p(\boldsymbol{\Gamma}) \propto \prod_{k=1}^M \underbrace{\exp\left(-\lambda \|\sigma_\alpha(\boldsymbol{\Gamma}_k)\|_1\right)}_{\text{sparse masks}}$$
$$\cdot \prod_{i=1}^d \underbrace{\text{Beta}(\sigma_\alpha(\gamma_{k,i}); \zeta_1, \zeta_2)}_{\text{sharp masks}} \underbrace{\mathcal{N}(\gamma_k|\mathbf{0}, \eta_\gamma^2 \mathbf{I})}_{\text{numerical stability}} \tag{8}$$

We assume that interventions occur only infrequently, i.e., in expectation only on one variable. Hence, we choose $\zeta_1 = 1/d$ and $\zeta_2 = (d-1)/d$. The sparsity prior implies that, given an active intervention target $i$, it is a-priori less likely that a variable $j \neq i$ is intervened upon in the same context $k$.

**Interventional likelihood.** We obtain a differentiable version of the interventional likelihood by sampling masks $\mathcal{I}^{\text{tar}} \sim \text{Bernoulli}(\sigma_\alpha(\boldsymbol{\Gamma}))$ with the Gumbel-Softmax trick [Jang et al., 2016, Maddison et al., 2017] and using them as a switch between observational and interventional likelihoods per variable:

$$p(\mathcal{D}_k|\mathbf{G}, \boldsymbol{\Theta}, I_k^{\text{tar}}, \boldsymbol{\Theta}_{I_k}) = \tag{9}$$
$$\prod_{j=1}^{n_k} \prod_{i=1}^d \left( p(x_i^{(k,j)}|x_{\text{pa}_{\mathbf{G}}(i)}, \boldsymbol{\Theta})^{(1-I_{k,i}^{\text{tar}})} \cdot p(x_i^{(k,j)}|\boldsymbol{\Theta}_{I_k})^{I_{k,i}^{\text{tar}}} \right)$$

**Stein Variational Gradient Descent.** Having rewritten the inference problem to continuous latent variables $\mathbf{Z}$ and $\boldsymbol{\Gamma}$, the final challenge is approximating the intractable posterior $p(\mathbf{Z}, \boldsymbol{\Theta}, \boldsymbol{\Gamma}, \boldsymbol{\Theta}_{\mathcal{I}}|\mathbf{D})$. To this end, we employ the particle variational inference approach SVGD [Liu and Wang, 2016], which minimizes the KL divergence to the intractable distribution of interest using a finite set of particles. It relies on the score of the distribution to guide the particles towards regions of high probability while using a kernel function $k(\cdot, \cdot)$ between them. We employ a sum of RBF kernels as the kernel function over $\boldsymbol{\Psi} := (\mathbf{Z}, \boldsymbol{\Theta}, \boldsymbol{\Gamma}, \boldsymbol{\Theta}_{\mathcal{I}})$.

The proposed algorithm is summarized in Alg. 1 of Appx. B. There, we also give details on how to derive the scores of the log-likelihood and the chosen kernel. Figure 4 in Appx. B illustrates an example of the returned posterior particles for $\mathbf{G}$ and $\mathcal{I}_{\text{tar}}$ alongside the ground truth for the case of a linear Gaussian 5 node graph.

# 6 EXPERIMENTS

We evaluate BaCaDI on different causal discovery tasks with data from multiple contexts. Our aim is to empirically investigate how accurately BaCaDI predicts the causal structure as well as the intervention targets and how it compares to related state-of-the-art methods. First, we focus on synthetic datasets. Second, we use the SERGIO simulator [Dibaeinia and Sinha, 2020] to evaluate the methods on a more realistic task of simulated gene expression data. Finally, we analyse how well the uncertainty estimates are calibrated.

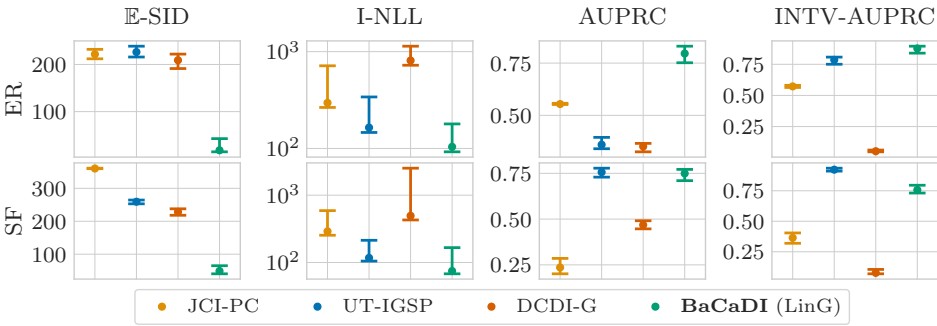

Figure 2: Joint post. inference over CBNs and interventions for *linear Gaussian* ground-truth CBNs. The results are for data from ER-2 and SF-2 graphs with $d = 20$ variables and $M = 20$ contexts.

## 6.1 EXPERIMENT METHODOLOGY

**Datasets.** Following related work [Zheng et al., 2018, Yu et al., 2019, Zheng et al., 2020, Annadani et al., 2021, Scherrer et al., 2021, Lorch et al., 2021], we perform inference of random graphs with different local conditionals. We consider Erdős-Rényi [Gilbert, 1959] and scale-free [Barabási and Albert, 1999] random graphs with $d = 20$ nodes and $2d$ edges in expectation (ER-2 and SF-2). We create datasets by randomly sampling parameters or simulating data with SERGIO. We then split the data into training and held-out test datasets. In all settings, we collect $n_0 = 100$ observational samples together with $n_k = 10$ samples per intervention context for $k \in \{1, \dots, M\}$. We defer further details on data generation to Appx. D.

**Baselines.** We compare BaCaDI to existing algorithms that are capable of joint causal inference from multiple contexts with unknown interventions. This includes the constraint-based methods UT-IGSP [Squires et al., 2020] and the JCI framework with the PC algorithm (JCI-PC) [Mooij et al., 2016], and the score-based method DCDI [Brouillard et al., 2020] that can handle unknown interventions. JCI-PC and UT-IGSP are based on conditional independence or invariance tests. For DCDI, we use a neural network to model the local conditionals with Gaussian additive noise (DCDI-G). This is the same model class and capacity as BaCaDI with a nonlinear likelihood model. Since all of these methods arrive only at a single DAG estimate, we use the non-parametric DAG bootstrap approach [Friedman et al., 2013, Agrawal et al., 2019] to obtain an approximate distribution over DAGs and intervention targets. Throughout the experiments, 20 bootstrap samples are used.

**BaCaDI instantiations.** We instantiate BaCaDI using 20 particles for SVGD and run it for 2000 steps. Unless specified otherwise, we model interventions using a Gaussian likelihood $p_i^{I_k}(x_i | \boldsymbol{\Theta}_{I_k}) = \mathcal{N}(x_i | \mu_{k,i}^I, \sigma_I)$ with fixed variance $\sigma_I^2 = 0.5$. We infer the means $\boldsymbol{\Theta}_{I_k} = [\mu_{k,1}^I, \dots, \mu_{k,d}^I]$ using a wide Gaussian prior $p(\boldsymbol{\Theta}_{I_k} | I_k^{\text{tar}}) = \prod_{i \in I_k^{\text{tar}}} p(\mu_{k,i}^I | I_{k,i}^{\text{tar}} = 1) = \prod_{i \in I_k^{\text{tar}}} \mathcal{N}(\mu_{k,i}^I | 0, 10)$. This reflects an uninformative prior over a large effect

range of the interventions. The local conditionals for the observational likelihood are either modelled as linear or nonlinear models with additive Gaussian noise. The latter uses 1 hidden-layer neural networks (NN) with 5 hidden units. We use 20 particles to instantiate the SVGD algorithm. More details about the models can be found in Appx. D.1.

**Metrics.** Our reported metrics focus on the essential aspects of our inference problem: causal graph prediction, intervention detection, and inference of the full CBN conditionals.

*Causal Discovery:* We focus on the *Structural Interventional Distance* (SID) [Peters and Bühlmann, 2015] that quantifies the closeness between two DAGs in terms of how well their interventional adjustment sets coincide. Since we perform posterior inference, we consider the *expected* SID: $\mathbb{E}\text{-SID}(p, \mathbf{G}_{\text{gt}}) := \sum_{\mathbf{G}} p(\mathbf{G} | \mathcal{D}) \cdot \text{SID}(\mathbf{G}, \mathbf{G}_{\text{gt}})$. Since UT-IGSP and JCI-PC only return a CPDAGs of the Interventional Markov Equivalence Class (I-MEC), we calculate its lower and upper bound of the SID, and report their midpoint as the $\mathbb{E}$-SID (see Appx. D.5). Furthermore, we compute the area under the precision recall curve (AUPRC) for pairwise edge prediction based on the posterior marginals $p(g_{ij} = 1 | \mathbf{D})$.

*Interventions*: We report *interventional* AUPRC (INTV-AUPRC) for the classification of targets.

*Learning conditionals:* We compute the average negative *interventional log-likelihood (I-NLL)* (see Appx. D.5) on $M^{\text{test}} = 10$ heldout interventional datasets. Since UT-IGSP and JCI-PC are not equipped with local conditional distributions, we use the linear Gaussian maximum-likelihood parameters (MLE) that are computed in closed-form [Hauser and Bühlmann, 2014] to compute the heldout I-NLL.

**Result aggregation.** For all methods and settings, we perform a search over a specified range of hyperparameters with at least 20 settings. For a specific choice of hyperparameters, we collect results over 30 different random graphs and pick the hyperparameters that resulted in the lowest I-NLL across the 30 instances in the held-out interventional dataset. We report the median of each metric together with its 90% confidence interval based on empirical percentiles.

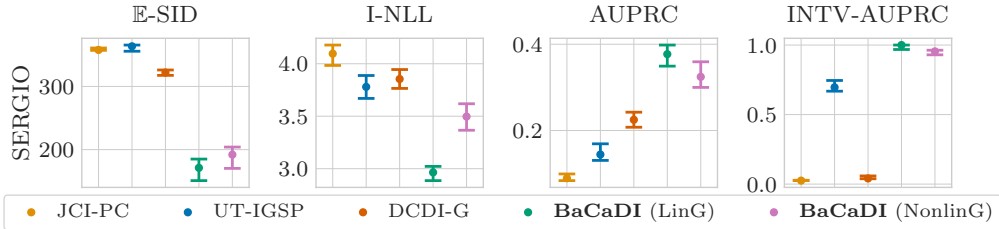

Figure 3: Joint posterior over CBNs and interventions for simulated *gene expression* data with $d = 20$ variables and $M = 10$ interv. contexts. BaCaDI makes significantly better causal mechanism predictions than the baselines and accurately identifies the correct intervention targets.

## 6.2 EXPERIMENT RESULTS ON SYNTHETIC DATA

First, we study the performance of BaCaDI and the baselines on joint causal inference from synthetic data. We focus on hard interventions that set the value of the targeted variable to a Gaussian with a randomly chosen mean bounded away from zero. Each interventional dataset is generated by intervening on a specific variable. We target all variables in the graph, i.e., the number of interventional contexts is $M = d$. In total, this results in 300 samples for synthetic datasets with $d = 20$ variables. We first consider *linear Gaussian* CBNs where each variable is a linear combination of its parents with additive Gaussian noise. The corresponding results for ER-2 and SF-2 are shown in Fig. 2.

Across the four synthetic evaluation settings, BaCaDI's causal structure predictions are the closest to the ground-truth CBN a) in terms of their intervention implications (i.e., $\mathbb{E}$-SID, I-NLL) as well as b) for predicting individual edges (i.e., AUPRC). In most cases, our method outperforms the baselines by a significant margin. Moreover, BaCaDI achieves strong AUPRC scores for the prediction of intervention targets. Among the baselines, UT-IGSP is the best at detecting interventions, largely on par with BaCaDI. Although UT-IGSP also achieves high AUPRC for the linear SF-2 setting, the method significantly falls behind BaCaDI in other settings and metrics.

## 6.3 EXPERIMENTS ON SIMULATED GENE-REGULATORY NETWORKS

We evaluate all methods in a realistic application domain using SERGIO [Dibaeinia and Sinha, 2020], a simulator for single-cell expression data of gene regulatory networks. Given a user-defined causal graph **G**, SERGIO utilizes stochastic differential equations to simulate the gene expression dynamics and generate realistic single-cell transcriptomic datasets, which correspond to samples from the steady state of this dynamical system. Since real-world gene regulatory networks resemble scale-free structures [Albert, 2005, Ouma et al., 2018], we use randomly sampled SF-2 graphs with $d = 20$. In this domain, we perform $M = 10$ gene knockout interventions on a single randomly selected

target per context, resulting in a dataset size of 200 including the observational samples. See Appx. D.3 for more details.

Our Bayesian formulation allows us to embed prior knowledge into the inference process in a principled manner. Since we perform knockout interventions, we expect intervention values to be close to zero and exhibit low variance. To reflect this prior belief, we set the intervention noise to $\sigma_I^2 = 0.01$ and use the prior $p(\mu_{k,i}^I | I_{k,i}^{\text{tar}} = 1) = \mathcal{N}(\mu_{k,i}^I | 0, 1)$.

**Accuracy.** Fig. 3 shows the results for the SERGIO datasets. As in the synthetic CBN domain, BaCaDI infers the ground-truth graph most accurately given the provided data. Moreover, it is very accurate in predicting intervention targets as reflected by the intervention target AUPRC score, where it benefits from the prior.

**Uncertainty.** We utilize the concept of *calibration* [Gneiting et al., 2007, Kuleshov et al., 2018] to quantify the reliability of our uncertainty estimates. The results are shown in Appx. E.3. Notably, BaCaDI is the only method that takes into account the *epistemic uncertainty* when dealing with limited data and shows that its probabilistic predictions are reliable.

## 7 CONCLUSION

We introduce BaCaDI, a fully-differentiable Bayesian causal discovery framework for data generated under various unknown experimental/interventional conditions. BaCaDI performs approximate inference jointly over the underlying graph, mechanisms, and unknown interventions. A key feature is its principled treatment of epistemic uncertainty which allows it to work reliably even when data is scarce, providing well-calibrated uncertainty estimates alongside its structural predictions. Our work is motivated by the challenging problem of inferring the causal mechanisms of gene regulatory networks from real single-cell gene expression data. Our experimental results for the simulated gene expression data show that BaCaDI brings us one step closer to this ambitious goal. To ultimately reach it, a range of further challenges, such as dealing with the experimental measurement noise incurred by single-cell sequencing techniques, will have to be overcome by future work.

**Acknowledgements**

This research was supported by the European Research Council (ERC) under the European Union's Horizon 2020 research and innovation program grant agreement no. 815943 and the Swiss National Science Foundation under NCCR Automation, grant agreement 51NF40 180545. Jonas Rothfuss was supported by an Apple Scholars in AI/ML fellowship. We thank Parnian Kassraie for valuable feedback.

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

# A PROOFS

## A.1 LATENT POSTERIOR EXPECTATION

Recall that we have the generative model under which we factorize

$$p(\mathbf{Z}, \mathbf{G}, \boldsymbol{\Theta}, \boldsymbol{\Gamma}, \mathcal{I}, \mathbf{D}) = \underbrace{p(\mathbf{Z})p(\mathbf{G}|\mathbf{Z})p(\boldsymbol{\Theta}|\mathbf{G})}_{\text{gen. process CBN}} \prod_{k=1}^{M} \underbrace{p(\boldsymbol{\Gamma}_k)p(I_k^{\text{tar}}|\boldsymbol{\Gamma}_k)p(\boldsymbol{\Theta}_{I_k}|I_k^{\text{tar}})}_{\text{gen. process interv.}} \underbrace{p(\mathcal{D}_k|\mathbf{G}, \boldsymbol{\Theta}, I_k^{\text{tar}}, \boldsymbol{\Theta}_{I_k})}_{\text{interv. likelihood}}$$

where we write $\boldsymbol{\Gamma} := [\boldsymbol{\Gamma}_1, ..., \boldsymbol{\Gamma}_M]$. For brevity, we can express this as

$$p(\mathbf{Z}, \mathbf{G}, \boldsymbol{\Theta}, \boldsymbol{\Gamma}, \mathcal{I}, \mathbf{D}) = p(\mathbf{Z})p(\mathbf{G}|\mathbf{Z})p(\boldsymbol{\Theta}|\mathbf{G}) \cdot p(\boldsymbol{\Gamma})p(\mathcal{I}^{\text{tar}}|\boldsymbol{\Gamma})p(\boldsymbol{\Theta}_{\mathcal{I}}|\mathcal{I}^{\text{tar}}) \cdot p(\mathbf{D}|\mathbf{G}, \mathcal{I}, \boldsymbol{\Theta})$$

where $p(\mathbf{D}|\mathbf{G}, \mathcal{I}, \boldsymbol{\Theta}) = p(\mathbf{D}|\mathbf{G}, \boldsymbol{\Theta}, \mathcal{I}^{\text{tar}}, \boldsymbol{\Theta}_{\mathcal{I}}) = \prod_{k=1}^{M} p(\mathcal{D}_k|\mathbf{G}, \boldsymbol{\Theta}, I_k^{\text{tar}}, \boldsymbol{\Theta}_{I_k})$.

This gives us

$$\mathbb{E}_{p(\mathbf{G}, \boldsymbol{\Theta}, \mathcal{I}|\mathbf{D})}[f(\mathbf{G}, \boldsymbol{\Theta}, \mathcal{I})]$$

$$= \sum_{\mathbf{G}} \int_{\boldsymbol{\Theta}} \sum_{\mathcal{I}^{\text{tar}}} \int_{\boldsymbol{\Theta}_{\mathcal{I}}} p(\mathbf{G}, \boldsymbol{\Theta}, \mathcal{I}|\mathbf{D})f(\mathbf{G}, \boldsymbol{\Theta}, \mathcal{I})d\boldsymbol{\Theta}d\boldsymbol{\Theta}_{\mathcal{I}}$$

$$= \sum_{\mathbf{G}} \int_{\boldsymbol{\Theta}} \sum_{\mathcal{I}^{\text{tar}}} \int_{\boldsymbol{\Theta}_{\mathcal{I}}} p(\mathbf{G}, \mathcal{I}^{\text{tar}}, \boldsymbol{\Theta}, \boldsymbol{\Theta}_{\mathcal{I}}|\mathbf{D})f(\mathbf{G}, \boldsymbol{\Theta}, \mathcal{I})d\boldsymbol{\Theta}d\boldsymbol{\Theta}_{\mathcal{I}}$$

(splitting $\mathcal{I}$ to $\mathcal{I}^{\text{tar}}, \boldsymbol{\Theta}_{\mathcal{I}}$)

$$= \sum_{\mathbf{G}} \int_{\boldsymbol{\Theta}} \sum_{\mathcal{I}^{\text{tar}}} \int_{\boldsymbol{\Theta}_{\mathcal{I}}} \frac{p(\mathbf{G}, \mathcal{I}^{\text{tar}}, \boldsymbol{\Theta}, \boldsymbol{\Theta}_{\mathcal{I}}, \mathbf{D})f(\mathbf{G}, \boldsymbol{\Theta}, \mathcal{I})}{p(\mathbf{D})}d\boldsymbol{\Theta}d\boldsymbol{\Theta}_{\mathcal{I}}$$

$$= \sum_{\mathbf{G}} \int_{\boldsymbol{\Theta}} \sum_{\mathcal{I}^{\text{tar}}} \int_{\boldsymbol{\Theta}_{\mathcal{I}}} \int_{\mathbf{Z}} \int_{\boldsymbol{\Gamma}} \frac{p(\mathbf{Z}, \mathbf{G}, \boldsymbol{\Gamma}, \mathcal{I}^{\text{tar}}, \boldsymbol{\Theta}, \boldsymbol{\Theta}_{\mathcal{I}}, \mathbf{D})f(\mathbf{G}, \boldsymbol{\Theta}, \mathcal{I})}{p(\mathbf{D})}d\mathbf{Z}d\boldsymbol{\Gamma}d\boldsymbol{\Theta}d\boldsymbol{\Theta}_{\mathcal{I}}$$

(extending by $\mathbf{Z}, \boldsymbol{\Gamma}$)

$$= \int_{\mathbf{Z}, \boldsymbol{\Gamma}, \boldsymbol{\Theta}, \boldsymbol{\Theta}_{\mathcal{I}}} \sum_{\mathbf{G}, \mathcal{I}^{\text{tar}}} \frac{p(\mathbf{Z}, \mathbf{G}, \boldsymbol{\Gamma}, \mathcal{I}^{\text{tar}}, \boldsymbol{\Theta}, \boldsymbol{\Theta}_{\mathcal{I}}, \mathbf{D})f(\mathbf{G}, \boldsymbol{\Theta}, \mathcal{I})}{p(\mathbf{D})}d\mathbf{Z}d\boldsymbol{\Gamma}d\boldsymbol{\Theta}d\boldsymbol{\Theta}_{\mathcal{I}}$$

(rearranging)

$$= \int_{\mathbf{Z}, \boldsymbol{\Gamma}, \boldsymbol{\Theta}, \boldsymbol{\Theta}_{\mathcal{I}}} \sum_{\mathbf{G}, \mathcal{I}^{\text{tar}}} \frac{p(\mathbf{Z})p(\mathbf{G}|\mathbf{Z})p(\boldsymbol{\Theta}|\mathbf{G})p(\boldsymbol{\Gamma})p(\mathcal{I}^{\text{tar}}|\boldsymbol{\Gamma})p(\boldsymbol{\Theta}_{\mathcal{I}}|\mathcal{I}^{\text{tar}})p(\mathbf{D}|\mathbf{G}, \mathcal{I}, \boldsymbol{\Theta})f(\mathbf{G}, \boldsymbol{\Theta}, \mathcal{I})}{p(\mathbf{D})}d\mathbf{Z}d\boldsymbol{\Gamma}d\boldsymbol{\Theta}d\boldsymbol{\Theta}_{\mathcal{I}}$$

(by the generative model)

$$= E_{p(\mathbf{Z}, \boldsymbol{\Theta}, \boldsymbol{\Gamma}, \boldsymbol{\Theta}_{\mathcal{I}}|\mathbf{D})} \left[ \sum_{\mathbf{G}, \mathcal{I}^{\text{tar}}} \frac{p(\mathbf{G}|\mathbf{Z})p(\boldsymbol{\Theta}|\mathbf{G})p(\mathcal{I}^{\text{tar}}|\boldsymbol{\Gamma})p(\boldsymbol{\Theta}_{\mathcal{I}}|\mathcal{I}^{\text{tar}})p(\mathbf{D}|\mathbf{G}, \mathcal{I}, \boldsymbol{\Theta})f(\mathbf{G}, \boldsymbol{\Theta}, \mathcal{I})}{p(\boldsymbol{\Theta}, \boldsymbol{\Theta}_{\mathcal{I}}, \mathbf{D}|\mathbf{Z}, \boldsymbol{\Gamma})} \right]$$

(since $p(\mathbf{Z}, \boldsymbol{\Theta}, \boldsymbol{\Gamma}, \boldsymbol{\Theta}_{\mathcal{I}}|\mathbf{D}) = \frac{p(\mathbf{Z})p(\boldsymbol{\Gamma})p(\boldsymbol{\Theta}, \boldsymbol{\Theta}_{\mathcal{I}}, \mathbf{D}|\mathbf{Z}, \boldsymbol{\Gamma})}{p(\mathbf{D})}$)

$$= E_{p(\mathbf{Z}, \boldsymbol{\Theta}, \boldsymbol{\Gamma}, \boldsymbol{\Theta}_{\mathcal{I}}|\mathbf{D})} \left[ \frac{\sum_{\mathbf{G}, \mathcal{I}^{\text{tar}}} p(\mathbf{G}|\mathbf{Z})p(\boldsymbol{\Theta}|\mathbf{G})p(\mathcal{I}^{\text{tar}}|\boldsymbol{\Gamma})p(\boldsymbol{\Theta}_{\mathcal{I}}|\mathcal{I}^{\text{tar}})p(\mathbf{D}|\mathbf{G}, \mathcal{I}, \boldsymbol{\Theta})f(\mathbf{G}, \boldsymbol{\Theta}, \mathcal{I})}{p(\boldsymbol{\Theta}, \boldsymbol{\Theta}_{\mathcal{I}}, \mathbf{D}|\mathbf{Z}, \boldsymbol{\Gamma})} \right]$$

(rearranging )

$$= E_{p(\mathbf{Z}, \boldsymbol{\Theta}, \boldsymbol{\Gamma}, \boldsymbol{\Theta}_{\mathcal{I}}|\mathbf{D})} \left[ \frac{\sum_{\mathbf{G}, \mathcal{I}^{\text{tar}}} p(\mathbf{G}|\mathbf{Z})p(\boldsymbol{\Theta}|\mathbf{G})p(\mathcal{I}^{\text{tar}}|\boldsymbol{\Gamma})p(\boldsymbol{\Theta}_{\mathcal{I}}|\mathcal{I}^{\text{tar}})p(\mathbf{D}|\mathbf{G}, \mathcal{I}, \boldsymbol{\Theta})f(\mathbf{G}, \boldsymbol{\Theta}, \mathcal{I})}{\sum_{\mathbf{G}, \mathcal{I}^{\text{tar}}} p(\mathbf{G}, \mathcal{I}^{\text{tar}}, \boldsymbol{\Theta}, \boldsymbol{\Theta}_{\mathcal{I}}, \mathbf{D}|\mathbf{Z}, \boldsymbol{\Gamma})} \right]$$

(law of total probability )

$$= \mathbb{E}_{p(\mathbf{Z}, \boldsymbol{\Theta}, \boldsymbol{\Gamma}, \boldsymbol{\Theta}_{\mathcal{I}}|\mathbf{D})} \left[ \frac{\sum_{\mathbf{G}, \mathcal{I}^{\text{tar}}} p(\mathbf{G}|\mathbf{Z})p(\boldsymbol{\Theta}|\mathbf{G})p(\mathcal{I}^{\text{tar}}|\boldsymbol{\Gamma})p(\boldsymbol{\Theta}_{\mathcal{I}}|\mathcal{I}^{\text{tar}})p(\mathbf{D}|\mathbf{G}, \mathcal{I}, \boldsymbol{\Theta})f(\mathbf{G}, \boldsymbol{\Theta}, \mathcal{I})}{\sum_{\mathbf{G}, \mathcal{I}^{\text{tar}}} p(\mathbf{G}|\mathbf{Z})p(\boldsymbol{\Theta}|\mathbf{G})p(\mathcal{I}^{\text{tar}}|\boldsymbol{\Gamma})p(\boldsymbol{\Theta}_{\mathcal{I}}|\mathcal{I}^{\text{tar}})p(\mathbf{D}|\mathbf{G}, \mathcal{I}, \boldsymbol{\Theta})} \right]$$

(by the generative model )

$$= \mathbb{E}_{p(\mathbf{Z},\mathbf{\Theta},\mathbf{\Gamma},\mathbf{\Theta}_{\mathcal{I}}|\mathbf{D})} \left[ \frac{\mathbb{E}_{p(\mathbf{G}|\mathbf{Z}),p(\mathcal{I}^{\mathrm{tar}}|\mathbf{\Gamma})}[f(\mathbf{G},\mathbf{\Theta},\mathcal{I})p(\mathbf{\Theta}|\mathbf{G})p(\mathbf{\Theta}_{\mathcal{I}}|\mathcal{I}^{\mathrm{tar}})p(\mathbf{D}|\mathbf{G},\mathcal{I},\mathbf{\Theta})]}{\mathbb{E}_{p(\mathbf{G}|\mathbf{Z}),p(\mathcal{I}^{\mathrm{tar}}|\mathbf{\Gamma})}[p(\mathbf{\Theta}|\mathbf{G})p(\mathbf{\Theta}_{\mathcal{I}}|\mathcal{I}^{\mathrm{tar}})p(\mathbf{D}|\mathbf{G},\mathcal{I},\mathbf{\Theta})]} \right]$$

## A.2 SCORES

We will derive the gradients of the unnormalized posterior since

$$\nabla_{\mathbf{Z}} \log p(\mathbf{Z},\mathbf{\Gamma},\mathbf{\Theta},\mathbf{\Theta}_{\mathcal{I}}|\mathcal{D}) = \nabla_{\mathbf{Z}} \log p(\mathbf{Z},\mathbf{\Gamma},\mathbf{\Theta},\mathbf{\Theta}_{\mathcal{I}},\mathcal{D}) - \nabla_{\mathbf{Z}} \log p(\mathcal{D}) \tag{10}$$

$$= \nabla_{\mathbf{Z}} \log p(\mathbf{Z},\mathbf{\Gamma},\mathbf{\Theta},\mathbf{\Theta}_{\mathcal{I}},\mathcal{D}) \tag{11}$$

and analogously for the gradients w.r.t. to other variables $\mathbf{Z},\mathbf{\Gamma},\mathbf{\Theta},\mathbf{\Theta}_{\mathcal{I}}$ .

By basic rules of probability theory and using the identity $\nabla_{\mathbf{x}} \log p(\mathbf{x}) = \nabla_{\mathbf{x}} p(\mathbf{x})/p(\mathbf{x})$, we obtain

$$\nabla_{\mathbf{Z}} \log p(\mathbf{Z},\mathbf{\Gamma},\mathbf{\Theta},\mathbf{\Theta}_{\mathcal{I}},\mathbf{D}) = \tag{12}$$

$$= \nabla_{\mathbf{Z}} \log p(\mathbf{Z}) + \nabla_{\mathbf{Z}} \log p(\mathbf{\Theta},\mathbf{\Theta}_{\mathcal{I}},\mathbf{D}|\mathbf{Z},\mathbf{\Gamma}) \tag{13}$$

$$= \nabla_{\mathbf{Z}} \log p(\mathbf{Z}) + \frac{\nabla_{\mathbf{Z}} p(\mathbf{\Theta},\mathbf{\Theta}_{\mathcal{I}},\mathbf{D}|\mathbf{Z},\mathbf{\Gamma})}{p(\mathbf{\Theta},\mathbf{\Theta}_{\mathcal{I}},\mathbf{D}|\mathbf{Z},\mathbf{\Gamma})} \tag{14}$$

$$= \nabla_{\mathbf{Z}} \log p(\mathbf{Z}) + \frac{\nabla_{\mathbf{Z}} \left[ \sum_{\mathbf{G}} \sum_{\mathcal{I}^{\mathrm{tar}}} p(\mathbf{G}|\mathbf{Z})p(\mathcal{I}^{\mathrm{tar}}|\mathbf{\Gamma})p(\mathbf{\Theta},\mathbf{\Theta}_{\mathcal{I}},\mathbf{D}|\mathbf{G},\mathcal{I}^{\mathrm{tar}}) \right]}{\sum_{\mathbf{G}} \sum_{\mathcal{I}^{\mathrm{tar}}} p(\mathbf{G}|\mathbf{Z})p(\mathcal{I}^{\mathrm{tar}}|\mathbf{\Gamma})p(\mathbf{\Theta},\mathbf{\Theta}_{\mathcal{I}},\mathbf{D}|\mathbf{G},\mathcal{I}^{\mathrm{tar}})} \tag{15}$$

$$= \nabla_{\mathbf{Z}} \log p(\mathbf{Z}) + \frac{\nabla_{\mathbf{Z}} \mathbb{E}_{p(\mathbf{G}|\mathbf{Z})} \mathbb{E}_{p(\mathcal{I}^{\mathrm{tar}}|\mathbf{\Gamma})} \left[ p(\mathbf{\Theta},\mathbf{\Theta}_{\mathcal{I}},\mathbf{D}|\mathbf{G},\mathcal{I}^{\mathrm{tar}}) \right]}{\mathbb{E}_{p(\mathbf{G}|\mathbf{Z})} \mathbb{E}_{p(\mathcal{I}^{\mathrm{tar}}|\mathbf{\Gamma})} \left[ p(\mathbf{\Theta},\mathbf{\Theta}_{\mathcal{I}},\mathbf{D}|\mathbf{G},\mathcal{I}^{\mathrm{tar}}) \right]} \tag{16}$$

and analogously

$$\nabla_{\mathbf{\Gamma}} \log p(\mathbf{Z},\mathbf{\Gamma},\mathbf{\Theta},\mathbf{\Theta}_{\mathcal{I}},\mathbf{D}) = \tag{17}$$

$$= \nabla_{\mathbf{\Gamma}} \log p(\mathbf{\Gamma}) + \nabla_{\mathbf{\Gamma}} \log p(\mathbf{\Theta},\mathbf{\Theta}_{\mathcal{I}},\mathbf{D}|\mathbf{Z},\mathbf{\Gamma}) \tag{18}$$

$$= \nabla_{\mathbf{\Gamma}} \log p\mathbf{\Gamma}) + \frac{\nabla_{\mathbf{\Gamma}} p(\mathbf{\Theta},\mathbf{\Theta}_{\mathcal{I}},\mathbf{D}|\mathbf{Z},\mathbf{\Gamma})}{p(\mathbf{\Theta},\mathbf{\Theta}_{\mathcal{I}},\mathbf{D}|\mathbf{Z},\mathbf{\Gamma})} \tag{19}$$

$$= \nabla_{\mathbf{\Gamma}} \log p(\mathbf{\Gamma}) + \frac{\nabla_{\mathbf{\Gamma}} \left[ \sum_{\mathbf{G}} \sum_{\mathcal{I}^{\mathrm{tar}}} p(\mathbf{G}|\mathbf{Z})p(\mathcal{I}^{\mathrm{tar}}|\mathbf{\Gamma})p(\mathbf{\Theta},\mathbf{\Theta}_{\mathcal{I}},\mathbf{D}|\mathbf{G},\mathcal{I}^{\mathrm{tar}}) \right]}{\sum_{\mathbf{G}} \sum_{\mathcal{I}^{\mathrm{tar}}} p(\mathbf{G}|\mathbf{Z})p(\mathcal{I}^{\mathrm{tar}}|\mathbf{\Gamma})p(\mathbf{\Theta},\mathbf{\Theta}_{\mathcal{I}},\mathbf{D}|\mathbf{G},\mathcal{I}^{\mathrm{tar}})} \tag{20}$$

$$= \nabla_{\mathbf{\Gamma}} \log p(\mathbf{\Gamma}) + \frac{\nabla_{\mathbf{\Gamma}} \mathbb{E}_{p(\mathbf{G}|\mathbf{z})} \mathbb{E}_{p(\mathcal{I}^{\mathrm{tar}}|\mathbf{\Gamma})} \left[ p(\mathbf{\Theta},\mathbf{\Theta}_{\mathcal{I}},\mathbf{D}|\mathbf{G},\mathcal{I}^{\mathrm{tar}}) \right]}{\mathbb{E}_{p(\mathbf{G}|\mathbf{z})} \mathbb{E}_{p(\mathcal{I}^{\mathrm{tar}}|\mathbf{\Gamma})} \left[ p(\mathbf{\Theta},\mathbf{\Theta}_{\mathcal{I}},\mathbf{D}|\mathbf{G},\mathcal{I}^{\mathrm{tar}}) \right]} . \tag{21}$$

# B ALGORITHM DETAILS

## B.1 BACKGROUND: BAYESIAN INFERENCE OF BNS

Given i.i.d. observations $\mathcal{D} = \{\mathbf{x}^{(1)},\ldots,\mathbf{x}^{(N)}\}$, Bayesian inference over BNs means constructing a *full posterior* probability density over BNs that model the observations. Following Friedman and Koller [2003], given a prior distribution over DAGs $p(\mathbf{G})$ and a prior over BN parameters $p(\mathbf{\Theta}|\mathbf{G})$, Bayes' Theorem yields the joint and marginal posterior distributions

$$p(\mathbf{G},\mathbf{\Theta}|\mathcal{D}) \propto p(\mathbf{G})p(\mathbf{\Theta}|\mathbf{G})p(\mathcal{D}|\mathbf{G},\mathbf{\Theta}) \tag{22}$$

$$p(\mathbf{G}|\mathcal{D}) \propto p(\mathbf{G})p(\mathcal{D}|\mathbf{G}) , \tag{23}$$

where $p(\mathcal{D}|\mathbf{G},\mathbf{\Theta}) = \prod_{i=1}^{n} p(\mathbf{x}^{(i)}|\mathbf{G},\mathbf{\Theta})$ the likelihood of the independent observations in $\mathcal{D}$ and $p(\mathcal{D}|\mathbf{G}) = \int p(\mathbf{\Theta}|\mathbf{G})p(\mathcal{D}|\mathbf{G},\mathbf{\Theta})d\mathbf{\Theta}$ is the marginal likelihood. Thus, $p(\mathcal{D}|\mathbf{G})$ in Eq. 23 is only available in closed form for special conjugate cases. The Bayesian formalism allows us to compute expectations of the form

$$\mathbb{E}_{p(\mathbf{G},\mathbf{\Theta}|\mathcal{D})} \Big[ f(\mathbf{G},\mathbf{\Theta}) \Big] \qquad \text{or} \qquad \mathbb{E}_{p(\mathbf{G}|\mathcal{D})} \Big[ f(\mathbf{G}) \Big] \tag{24}$$

for any function $f$ of interest. For instance, for Bayesian model averaging, we would use $f(\mathbf{G}, \boldsymbol{\Theta}) = p(\mathbf{x}|\mathbf{G}, \boldsymbol{\Theta})$ or $f(\mathbf{G}) = p(\mathbf{x}|\mathbf{G})$, respectively [Madigan and Raftery, 1994, Madigan et al., 1995]. In active learning of CBNs, a commonly used $f$ is the information gain in $\mathbf{G}$ from an intervention [Tong and Koller, 2001, Murphy, 2001, Cho et al., 2016, Agrawal et al., 2019]. Inferring the posterior is computationally challenging since there are $\mathcal{O}(d!2^{\binom{d}{2}})$ possible DAGs with $d$ nodes [Robinson, 1973]. Hence, computing the normalization constant $p(\mathcal{D})$ is also generally intractable.

## B.2 STEIN VARIATIONAL GRADIENT DESCENT OVER CBNS AND INTERVENTIONS

**Estimating Gradients of the Log-Posterior.** The score w.r.t. auxiliary variables $\mathbf{Z}$ and $\boldsymbol{\Gamma}$ requires marginalization over the corresponding discrete structures $\mathbf{G}$ and $\mathcal{I}^{\text{tar}}$; in particular

$$\nabla_{\mathbf{Z}} \log p(\mathbf{Z}, \boldsymbol{\Theta}, \boldsymbol{\Gamma}, \boldsymbol{\Theta}_{\mathcal{I}}|\mathbf{D}) = \nabla_{\mathbf{Z}} \log p(\mathbf{Z}) + \frac{\nabla_{\mathbf{Z}} \mathbb{E}_{p(\mathbf{G}|\mathbf{Z})} \mathbb{E}_{p(\mathcal{I}^{\text{tar}}|\boldsymbol{\Gamma})} [p(\boldsymbol{\Theta}, \boldsymbol{\Theta}_{\mathcal{I}}, \mathbf{D}|\mathbf{G}, \mathcal{I}^{\text{tar}})]}{\mathbb{E}_{p(\mathbf{G}|\mathbf{Z})} \mathbb{E}_{p(\mathcal{I}^{\text{tar}}|\boldsymbol{\Gamma})} [p(\boldsymbol{\Theta}, \boldsymbol{\Theta}_{\mathcal{I}}, \mathbf{D}|\mathbf{G}, \mathcal{I}^{\text{tar}})]} \quad (25)$$

$$\nabla_{\boldsymbol{\Gamma}} \log p(\mathbf{Z}, \boldsymbol{\Theta}, \boldsymbol{\Gamma}, \boldsymbol{\Theta}_{\mathcal{I}}|\mathbf{D}) = \nabla_{\boldsymbol{\Gamma}} \log p(\boldsymbol{\Gamma}) + \frac{\nabla_{\boldsymbol{\Gamma}} \mathbb{E}_{p(\mathbf{G}|\mathbf{Z})} \mathbb{E}_{p(\mathcal{I}^{\text{tar}}|\boldsymbol{\Gamma})} [p(\boldsymbol{\Theta}, \boldsymbol{\Theta}_{\mathcal{I}}, \mathbf{D}|\mathbf{G}, \mathcal{I}^{\text{tar}})]}{\mathbb{E}_{p(\mathbf{G}|\mathbf{Z})} \mathbb{E}_{p(\mathcal{I}^{\text{tar}}|\boldsymbol{\Gamma})} [p(\boldsymbol{\Theta}, \boldsymbol{\Theta}_{\mathcal{I}}, \mathbf{D}|\mathbf{G}, \mathcal{I}^{\text{tar}})]} \quad (26)$$

where $p(\boldsymbol{\Theta}, \boldsymbol{\Theta}_{\mathcal{I}}, \mathbf{D}|\mathbf{G}, \mathcal{I}^{\text{tar}}) = p(\boldsymbol{\Theta}|\mathbf{G}) \prod_{k=1}^{m} p(\mathcal{D}_k|\mathbf{G}, \boldsymbol{\Theta}, I_k^{\text{tar}}, \boldsymbol{\Theta}_{I_k}) p(\boldsymbol{\Theta}_{I_k}|I_k^{\text{tar}})$. The expectations in Eq. 25 and Eq. 26 can be estimated by sampling the intervention masks $\mathcal{I}^{\text{tar}} \sim \text{Bern}(\sigma_\alpha(\boldsymbol{\Gamma}))$ and the adjacency matrices $\mathbf{G} \sim \text{Bern}(\sigma_\alpha(\mathbf{U}\mathbf{V}^\top))$ in a differentiable manner with the Gumbel-Softmax trick [Jang et al., 2016, Maddison et al., 2017]. For obtaining a differentiable observational log-likelihood, we mask individual log-likelihood summands based on $\mathbf{G}$, as discussed in Sec 4.3 of Lorch et al. [2021]. In a similar fashion, we use the $\mathcal{I}^{\text{tar}}$ to switch between observational and interventional likelihoods per variable:

$$\log p(\mathcal{D}_k|\mathbf{G}, \boldsymbol{\Theta}, I_k^{\text{tar}}, \boldsymbol{\Theta}_{I_k}) = \sum_{j=1}^{n_k} \sum_{i=1}^{d} (1 - I_{k,i}^{\text{tar}}) \cdot \log p(x_i^{(k,j)}|x_{\text{pa}_{\mathbf{G}}(i)}, \boldsymbol{\Theta}) I_{k,i}^{\text{tar}} \cdot \log p(x_i^{(k,j)}|\boldsymbol{\Theta}_{I_k}))$$

**SVGD instantiation.** In Sec. 5 we have introduced a fully differentiable Bayesian model which translates the posterior inference over discrete graphs $\mathbf{G}$ and intervention targets $\mathcal{I}_{\text{tar}}$ into an inference problem over the continuous latent variables $\mathbf{Z}$ and $\boldsymbol{\Gamma}$. In this section, we discuss how to approach the final challenge of approximating the intractable posterior $p(\mathbf{Z}, \boldsymbol{\Theta}, \boldsymbol{\Gamma}, \boldsymbol{\Theta}_{\mathcal{I}}|\mathbf{D})$ over our continuous latent variables.

To this end, we employ the particle variational inference approach SVGD [Liu and Wang, 2016] which approximates the intractable distribution of interest by a finite set of particles. SVGD uses the score of the distribution to guide the particles towards regions of high probability while using a kernel function $k(\cdot, \cdot)$ between them. The latter introduces repulsive forces which make the particles disperse well across the domain. For a brief review of SVGD, please see Appx. F. We employ a sum of RBF kernels (details in Appx. B.4) as the kernel function over $\boldsymbol{\Psi} := (\mathbf{Z}, \boldsymbol{\Theta}, \boldsymbol{\Gamma}, \boldsymbol{\Theta}_{\mathcal{I}})$. We also considered a product of RBF kernels, though, found that the additive kernel composition performed better.

Starting with an initial set of $L$ particles $\{\boldsymbol{\Psi}_0^{(l)}\}_{l=1}^L = \{(\mathbf{Z}_0^{(l)}, \boldsymbol{\Gamma}_0^{(l)}, \boldsymbol{\Theta}_0^{(l)}, \boldsymbol{\Theta}_{\mathcal{I},0}^{(l)})\}_{l=1}^L$, we perform $T$ iterations of particle SVGD updates. Following Lorch et al. [2021], we use annealing schedules $\alpha_t \to \infty$ and $\beta_t \to \infty$ so that our continuous relaxations $\mathbf{G}_\alpha(\mathbf{Z})$ and $\mathcal{I}_\alpha^{\text{tar}}(\boldsymbol{\Gamma})$ converge to DAGs and actual sets of intervention targets (details in Appx B.3). We return $\{(\mathbf{G}_\infty(\mathbf{Z}_T^{(l)}), \boldsymbol{\Theta}_T^{(l)}, \mathcal{I}_\infty^{\text{tar}}(\boldsymbol{\Gamma}_T^{(l)}), \boldsymbol{\Theta}_{\mathcal{I},T}^{(l)})\}_{l=1}^L$ as the particle approximation of the posterior $p(\mathbf{G}, \boldsymbol{\Theta}, \mathcal{I}_{\text{tar}}, \boldsymbol{\Theta}_{\mathcal{I}}|\mathbf{D})$ with discrete DAGs and interventions targets. Figure 4 illustrates an example of the returned posterior particles for $\mathbf{G}$ and $\mathcal{I}_{\text{tar}}$ alongside the ground truth for the case of a linear Gaussian 5 node graph. The proposed algorithm is summarized in Alg. 1. While SVGD yields a set of particles of with equal weights, we weight each particle by its unnormalized posterior probability $p(\mathbf{G}, \boldsymbol{\Theta}, \boldsymbol{\Theta}_{\mathcal{I}}, \mathcal{I}^{\text{tar}}, |\mathbf{D})$ for performing approximate Bayesian model averaging. We find that this improves the empirical performance.

Although we employ SVGD, our general framework from Sec. 5 can also be instantiated with other score-based sampling [e.g. Welling and Teh, 2011, Chen et al., 2014] or variational inference methods [Blei et al., 2017]. Such alternatives may become favorable for systems of many variables, since the size of the adjacency matrix $\mathbf{G}$ grows quadratically in $d$ and the respective kernel component between $\mathbf{G}$'s becomes useless in high-dimensions.

## B.3 ANNEALING OF $\alpha$ AND $\beta$

The latent variables $\mathbf{Z}$ and $\boldsymbol{\Gamma}$ probabilistically model the causal graph $\mathbf{G}$ and intervention target masks $\mathcal{I}^{\text{tar}}$. In a similar way, they can be viewed as as continuous relaxations of $\mathbf{G}$ and $\mathcal{I}^{\text{tar}}$ respectively, where the $\alpha$ trades-off between smoothness

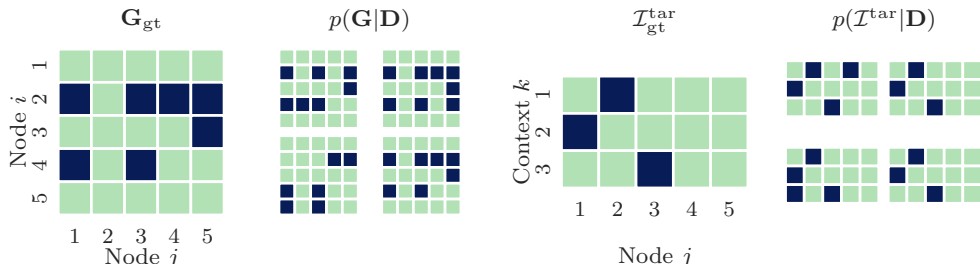

Figure 4: Instead of just one point estimate, BaCaDI yields particle approximations of the posteriors $p(\mathbf{G}|\mathbf{D})$ and $p(\mathcal{I}^{\text{tar}}|\mathbf{D})$. We visually compare the posterior particles with the ground truth $\mathbf{G}_{\text{gt}}$ and $\mathcal{I}^{\text{tar}}_{\text{gt}}$ for a SF-2 graph with $d = 5$ nodes and $M = 3$ contexts. Blue colors represent edges or targets.

and accuracy of these relaxations. If $\alpha \to \infty$, the sigmoid $\sigma_\alpha(\cdot)$ converges to the unit step function so that the probability distributions $p(\mathbf{G}|\mathbf{Z})$ and $p(\mathcal{I}^{\text{tar}}|\mathbf{\Gamma})$ become deterministic indicator functions, representing only single discrete graph $\mathbf{G}_\infty(\mathbf{Z}) \in \{0,1\}^{d \times d}$ and target mask $\mathcal{I}^{\text{tar}}_\infty(\mathbf{\Gamma}) \in \{0,1\}^{M \times d}$. To be able to make this simplification, as proposed in Lorch et al. [2021], we anneal $\alpha_t$ such that $\alpha_t \to \infty$ and, at terminal iteration of the SVGD training loop, convert $\mathbf{Z}$ and $\mathbf{\Gamma}$ into their discrete counterparts $\mathbf{G}_\infty(\mathbf{Z})$ and $\mathcal{I}^{\text{tar}}_\infty(\mathbf{\Gamma})$. Similarly, we set an annealing schedule $\beta_t \to \infty$ for inverse temperature parameter in the latent prior $p_\beta(\mathbf{Z})$ such that and we only model DAGs as the training progresses.

Importantly, this annealing process simplifies the expectation in Eq. 4 to

$$
\begin{aligned}
&\mathbb{E}_{p(\mathbf{G},\mathbf{\Theta},\mathcal{I}|\mathbf{D})}[f(\mathbf{G},\mathbf{\Theta},\mathcal{I})] \\
&= \mathbb{E}_{p(\mathbf{Z},\mathbf{\Theta},\mathbf{\Gamma},\mathbf{\Theta}_{\mathcal{I}}|\mathbf{D})}\left[ \frac{\mathbb{E}_{p(\mathbf{G}|\mathbf{Z}),p(\mathcal{I}^{\text{tar}}|\mathbf{\Gamma})}[f(\mathbf{G},\mathbf{\Theta},\mathcal{I})p(\mathbf{\Theta}|\mathbf{G})p(\mathbf{\Theta}_{\mathcal{I}}|\mathcal{I}^{\text{tar}})p(\mathbf{D}|\mathbf{G},\mathcal{I},\mathbf{\Theta})]}{\mathbb{E}_{p(\mathbf{G}|\mathbf{Z}),p(\mathcal{I}^{\text{tar}}|\mathbf{\Gamma})}[p(\mathbf{\Theta}|\mathbf{G})p(\mathbf{\Theta}_{\mathcal{I}}|\mathcal{I}^{\text{tar}})p(\mathbf{D}|\mathbf{G},\mathcal{I},\mathbf{\Theta})]} \right] \\
&\to \mathbb{E}_{p(\mathbf{Z},\mathbf{\Theta},\mathbf{\Gamma},\mathbf{\Theta}_{\mathcal{I}}|\mathbf{D})}[f(\mathbf{G}_\infty,\mathbf{\Theta},\mathcal{I}_\infty)] \,,
\end{aligned}
\tag{27}
$$

where $\mathcal{I}_\infty := (\mathcal{I}^{\text{tar}}_\infty(\mathbf{\Gamma}),\mathbf{\Theta}_{\mathcal{I}})$. This holds since the inner expectations evaluate to a single point for $\alpha_t,\beta_t \to \infty$ and thus cancel out.

### B.4 SVGD KERNEL

In our experiments, we employ an additive RBF kernel. We also considered a product of RBF kernels, though, found that the additive kernel composition performed better. This additive kernel is defined as

$$
\begin{aligned}
&k\left((\mathbf{Z},\mathbf{\Theta},\mathbf{\Gamma},\mathbf{\Theta}_{\mathcal{I}}),(\mathbf{Z}',\mathbf{\Theta}',\mathbf{\Gamma}',\mathbf{\Theta}'_{\mathcal{I}})\right) \\
&:= \exp\left(-\frac{\|\mathbf{Z}-\mathbf{Z}'\|^2}{2\tau_Z}\right) + \exp\left(-\frac{\|\mathbf{\Gamma}-\mathbf{\Gamma}'\|^2}{2\tau_\gamma}\right) \\
&+ \exp\left(-\frac{\|\mathbf{\Theta}-\mathbf{\Theta}'\|^2}{2\tau_\theta}\right) + \exp\left(-\frac{\|\mathbf{\Theta}_{\mathcal{I}}-\mathbf{\Theta}'_{\mathcal{I}}\|^2}{2\tau_\theta}\right)
\end{aligned}
\tag{28}
$$

with lengthscales $\tau_Z, \tau_\gamma, \tau_\theta$. For brevity, we also write $k(\mathbf{\Psi},\mathbf{\Psi}')$ for (28) where $\mathbf{\Psi} := (\mathbf{Z},\mathbf{\Theta},\mathbf{\Gamma},\mathbf{\Theta}_{\mathcal{I}})$. For SVGD, the kernel introduces repulsive forces which make the particles disperse well across the domain (see F). Importantly, the lengthscale hyperparameters provide the possibility to fine-tune the repulsion and hence calibrate our inference model. We show this in Sec. E.3.

### B.5 ALGORITHM OVERVIEW

We provide a pseudocode of our algorithm with its SVGD instantiation in Alg. 1.

## C MARGINAL INFERENCE WITH THE BGE SCORE

In addition to joint posterior inference that includes the *parameters* $\mathbf{\Theta}$ of the model, we also consider the *marginal* posterior $p(\mathbf{G}|\mathbf{D})$. We employ the commonly used Bayesian Gaussian Equivalent (BGe) marginal likelihood that scores Markov

---

**Algorithm 1** BaCaDI with SVGD for inference of $p(\mathbf{G}, \boldsymbol{\Theta}, \mathcal{I}^{\text{tar}}, \boldsymbol{\Theta}_{\mathcal{I}} | \mathbf{D})$

---

    **Input:** Set of datasets $\mathbf{D} = \{\mathcal{D}_0, ..., \mathcal{D}_M\}$ from the same causal system under different contexts
    **Input:** Kernel $k$, schedules for $\alpha_t, \beta_t$, and stepsizes $\eta_t$
    **Output:** Set of CBN and intervention particles $\{(\mathbf{G}^{(l)}, \boldsymbol{\Theta}^{(l)}, \mathcal{I}^{\text{tar},(l)}, \boldsymbol{\Theta}_{\mathcal{I}}^{(l)})\}_{l=1}^{L}$

1: Initialize set of latent and parameter particles $\{\boldsymbol{\Psi}_0^{(l)}\}_{l=1}^{L} = \{(\mathbf{Z}_0^{(l)}, \boldsymbol{\Gamma}_0^{(l)}, \boldsymbol{\Theta}_0^{(l)}, \boldsymbol{\Theta}_{\mathcal{I},0}^{(l)})\}_{l=1}^{L}$
2: **for** iteration $t = 0$ to $T - 1$ **do**
3:     Estimate $\nabla \log p(\mathbf{Z}, \boldsymbol{\Theta}, \boldsymbol{\Gamma}, \boldsymbol{\Theta}_{\mathcal{I}} | \mathbf{D})$ for each $\boldsymbol{\Psi}_t^{(l)} = (\mathbf{Z}_t^{(l)}, \boldsymbol{\Theta}_t^{(l)}, \boldsymbol{\Gamma}_t^{(l)}, \boldsymbol{\Theta}_{\mathcal{I},t}^{(l)})$         ▷ see Eq 25, 26
4:     **for** particle $m = l$ to $L$ **do**
5:         $\mathbf{Z}_{t+1}^{(l)} \leftarrow \mathbf{Z}_t^{(l)} + \eta_t \, \phi_t^{\mathbf{Z}}(\boldsymbol{\Psi}_t^{(l)})$         ▷ SVGD steps

        where $\phi_t^{\mathbf{Z}}(\cdot) := \frac{1}{L} \sum_{l=L}^{M} \left[ k\left(\boldsymbol{\Psi}_t^{(l)}, \cdot\right) \nabla_{\mathbf{Z}_t^{(l)}} \log p(\boldsymbol{\Psi}_t^{(l)} | \mathbf{D}) + \nabla_{\mathbf{Z}_t^{(l)}} k\left(\boldsymbol{\Psi}_t^{(l)}, \cdot\right) \right]$

6:         $\boldsymbol{\Theta}_{t+1}^{(l)} \leftarrow \boldsymbol{\Theta}_t^{(l)} + \eta_t \, \phi_t^{\boldsymbol{\Theta}}(\boldsymbol{\Psi}_t^{(l)})$
7:         $\boldsymbol{\Gamma}_{t+1}^{(l)} \leftarrow \boldsymbol{\Gamma}_t^{(l)} + \eta_t \, \phi_t^{\boldsymbol{\Gamma}}(\boldsymbol{\Psi}_t^{(l)})$
8:         $\boldsymbol{\Theta}_{\mathcal{I},t+1}^{(l)} \leftarrow \boldsymbol{\Theta}_{\mathcal{I},t}^{(l)} + \eta_t \, \phi_t^{\boldsymbol{\Theta}_{\mathcal{I}}}(\boldsymbol{\Psi}_t^{(l)})$

        where $\phi_t^{\boldsymbol{\Theta}}, \phi_t^{\boldsymbol{\Gamma}}, \phi_t^{\boldsymbol{\Theta}_{\mathcal{I}}}$ are analogous to $\phi_t^{\mathbf{Z}}$ but use gradients $\nabla_{\boldsymbol{\Theta}_t^{(l)}}, \nabla_{\boldsymbol{\Gamma}_t^{(l)}}, \nabla_{\boldsymbol{\Theta}_{\mathcal{I},t+1}^{(l)}}$

9: **return** $\{(\mathbf{G}_\infty(\mathbf{Z}_T^{(l)}), \boldsymbol{\Theta}_T^{(l)}, \mathcal{I}_\infty^{\text{tar}}(\boldsymbol{\Gamma}_T^{(l)}), \boldsymbol{\Theta}_{\mathcal{I},T}^{(l)})\}_{l=1}^{L}$         ▷ see Appx. B.3

---

equivalent structures equally [Geiger and Heckerman, 1994, 2002]. This model factorises the marginal likelihood into components for each node given its parents. Details on the computation of the BGe score are provided by Kuipers et al. [2014]. Other good explanations are given by Grzegorczyk [2010] and Kuipers and Moffa [2022].

Compared to the case of considering parameters $\boldsymbol{\Theta}$, the marginal likelihood does not yield factorization over different datasets. That is,

$$p(\mathbf{D} | \mathbf{G}, \mathcal{I}^{\text{tar}}) \neq \prod_{k=1}^{M} p(\mathcal{D}_k | \mathbf{G}, I_k^{\text{tar}})$$

Instead, we have to consider the *fused* data from all contexts by

$$\mathbf{X} = \begin{bmatrix} \mathbf{x}_1^T \\ \mathbf{x}_2^T \\ \vdots \\ \mathbf{x}_N^T \end{bmatrix} \in \mathbb{R}^{N \times d} \quad \text{with} \quad \mathbf{c} = \begin{bmatrix} c_1 \\ c_2 \\ \vdots \\ c_N \end{bmatrix} \in \{0, \ldots, M\}^N \tag{29}$$

for $N = \sum_{k=1}^{M} n_k$ and variables $c_i \in \{0, \ldots, M\}$ that indicate from which context sample $\mathbf{x}_i$ originates. As before, $c_i = 0$ denotes the observational context.

**Interventions.** When considering hard interventions, we cut off the connections of a variable to its parents. This effectively means that the data at this variable only contributes a constant (wrt. the current hypothesis graph) factor to the likelihood, and thus the scoring of a hypothesis graph should not be affected. In other words, when computing the score for a certain variable $j$ in the graph $\mathbf{G}$, we drop all datapoints in $\mathbf{X}$ where $j$ was the target of a hard intervention and then compute the BGe score for the remaining datapoints $\hat{\mathbf{X}}_j$. In our implementation, we achieve this by masking out datapoints in $\mathbf{X}$. This enables us to still make use of the Gumbel-Softmax estimator for the Bernoulli intervention targets.

**Priors.** Following the notation of Geiger and Heckerman [2002] and Kuipers et al. [2014], we use the standard effective sample size hyperparameters $\alpha_\mu = 1$ and $\alpha_\omega = d + 2$. Moreover, we use the diagonal form as the Wishart inverse scale matrix for the Normal-Wishart parameter prior underlying the BGe score (cf. [Grzegorczyk, 2010]) with a prior mean of $\boldsymbol{\mu} = [0, \ldots, 0]^T$.

For interventions, we use the same approach but set the prior values $\boldsymbol{\mu}$ to the real intervention mean from which the interventions where sampled. We then compute a BGe score for the intervened datapoints given an empty graph in order to have comparable likelihood. This helps when estimating the intervention targets that otherwise

**Experiments.** We evaluate BaCaDI using the marginal BGe score to the baselines of JCI-PC and UT-IGSP. Note that the algorithms of both JCI-PC and UT-IGSP are not affected by the different score; the difference is the scoring of the predicted

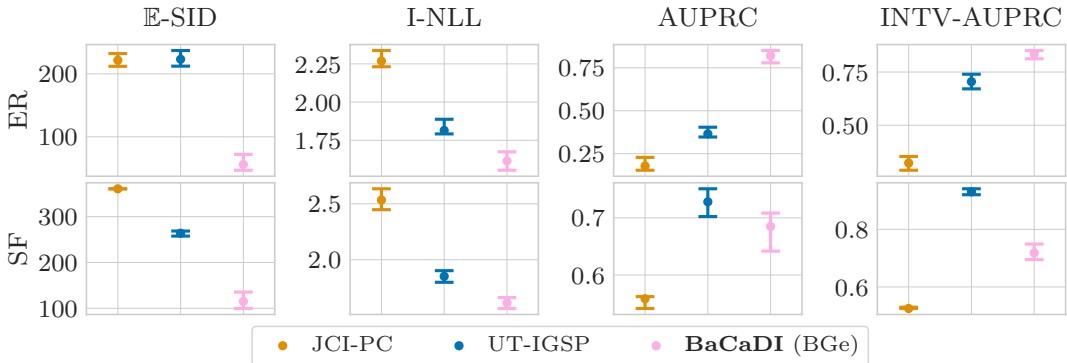

Figure 5: Additional results for Linear Gaussian BNs with $d = 20$ nodes. Here, we use the BGe score [Geiger and Heckerman, 1994, 2002] that marginalizes out the parameters of a linear Gaussian model. We again see how BaCaDI better predicts causal mechanisms across all settings. Note that DCDI-G is not included since it always performs joint inference with parameters.

graph structures, where we compute a likelihood score using the marginal BGe (replacing the MLE parameter estimate). We do not compare to DCDI-G as it always performs joint inference with parameters. Similar to Sec. 6, we focus on linear Gaussian BNs with $d = 20$ nodes. We report the results in Fig. 5. As done previously, we select the models with the lowest heldout I-NLL.

We again see how BaCaDI outperforms related methods in most metrics, particularly $\mathbb{E}$-SID and I-NLL. Only UT-IGSP performs competitively for AUPRC and INTV-AUPRC of SF graph structures, but fails to be on par with BaCaDI in other settings. These results resonate with the other experimental evaluations, showing promising results for making causal structure predictions.

## D EXPERIMENTAL SETUP

In the following, we describe the models, datasets and implementations that were used to perform experiments in detail.

### D.1 MODEL DETAILS & BACADI

First, we discuss the models of graphs as well as the exact usage of BaCaDI [1].

**Graphs.** We consider DAGs that follow either Erdős-Rényi (ER) [Gilbert, 1959] or scale-free (SF) [Barabási and Albert, 1999] distributions. ER graphs follow a prior distribution

$$p(\mathbf{G}) \propto q^{\|\mathbf{G}\|_1}(1 - q)^{\binom{d}{2} - \|\mathbf{G}\|_1} \tag{30}$$

that describes that each edge exists independently w.p. $q$. For SF graphs, we define a prior

$$p(\mathbf{G}) \propto \prod_{i=1}^{d}(1 + \|\mathbf{G}_i\|_1)^{-3} \tag{31}$$

analogous to their power law degree distribution $p(\text{deg}) \sim \text{deg}^{-3}$. $\mathbf{G}_i$ describes the $i$-th row of the adjacency matrix. For all our experiments, we use priors that result in $2d$ edges in expectation. Building on Lorch et al. [2021], BaCaDI can incorporate such a graph distribution in the prior $p(\mathbf{Z})$ (see Sec. 4.2 in their paper).

**Overview: Gaussian BNs.** As instantiations of BaCaDI 's inference models that describe the local conditional distributions, we consider Bayesian networks with additive Gaussian noise. This means that the variables $\mathbf{x} = (x_1, \ldots, x_d)$ follow a distribution

$$p(x_i|\mathbf{\Theta}, \mathbf{G}) = \mathcal{N}(f(x_{\text{pa}_{\mathbf{G}}(i)}, \mathbf{\Theta}), \sigma_i) \tag{32}$$

---

[1]Some parts of the model descriptions follow the appendix in [Lorch et al., 2021] and are included for completeness.

For the inference with BaCaDI , we assume a fixed observation noise $\sigma_i^2 = \sigma^2 = 0.1$.

The function $f$ can be modelled in different ways as follows.

**Linear BNs.** Linear Gaussian BNs model the mean of a given variable as a linear function of its parents:

$$p(\mathbf{x}|\mathbf{\Theta}, \mathbf{G}) = \mathcal{N}((\mathbf{G} \circ \mathbf{\Theta})^T \mathbf{x}, \sigma \mathbf{I}), \tag{33}$$

where $\circ$ denotes the element-wise multiplication. We use this parametrization as it allows for constant dimensionality of $\mathbf{\Theta} \in \mathbb{R}^{d \times d}$ and the elementwise multiplication only keeps the parents of each variable. Moreover, this allows the use of the Gumbel-Softmax estimator in Eq. 25.

As a prior, we use a standard Gaussian $p(\mathbf{\Theta}_{i,j}|\mathbf{G}_{i,j} = 1) = \mathcal{N}(0, 1)$.

**Nonlinear BNs.** The local conditionals can be extended to *nonlinear* dependencies modelled by neural networks. We consider feed-forward neural networks of the form

$$\text{FFN}(\mathbf{x}; \mathbf{\Theta}) := \mathbf{\Theta}^{(L)} \sigma(\ldots, \mathbf{\Theta}^{(1)} \mathbf{x} + \theta_b^{(1)}) \ldots) + \theta_b^{(L)}, \tag{34}$$

where $\mathbf{\Theta} = ((\mathbf{\Theta}^{(L)}, \theta_b^{(L)}), \ldots, (\mathbf{\Theta}^{(1)}, \theta_b^{(1)}))$ describe the parameters of the $l$-th layer for $l \in \{1, \ldots, L\}$. The function $\sigma$ is the element-wise nonlinear activation function and $\theta_b^{(l)}$ is the bias. This then gives the distribution

$$p(\mathbf{x}|\mathbf{\Theta}, \mathbf{G}) = \prod_{i=1}^{d} \mathcal{N}(x_i; \text{FFN}(\mathbf{G}_i^T \circ \mathbf{x}; \mathbf{\Theta}_i), \sigma). \tag{35}$$

Note that we thus have one neural network defined by $\mathbf{\Theta}_i$ for each of the local conditional distributions of variable $i$, that is, $d$ networks in total. For all experiments, we use one hidden layer with 5 units and the sigmoid activation function.

As a prior for the parameters, we analogously use a standard Gaussian with mean 0 and variance 1.

**Interventions.** We model interventions using a Gaussian likelihood $p_i^{I_k}(x_i|\mathbf{\Theta}_{I_k}) = \mathcal{N}(x_i|\mu_{k,i}^I, \sigma_I)$ with fixed variance $\sigma_I^2 = 0.5$. We infer the means $\mathbf{\Theta}_{I_k} = [\mu_{k,1}^I, \ldots, \mu_{k,d}^I]$ for which we set a wide Gaussian prior $p(\mathbf{\Theta}_{I_k}|I_k^{\text{tar}}) = \prod_{i \in I_k^{\text{tar}}} p(\mu_{k,i}^I|I_{k,i}^{\text{tar}} = 1) = \prod_{i \in I_k^{\text{tar}}} \mathcal{N}(\mu_{k,i}^I|0, 10)$. This reflects an uninformative prior over a large effect range of the interventions.

**Initialization.** For the linear Gaussians, the parameters are initialized closed to zero via $\mathbf{\Theta}_{\text{init}} \sim \mathcal{N}(0, \sigma_{\text{init}}\mathbf{I})$ with $\sigma_{\text{init}} = 0.3$. The closeness to zero is important to avoid inducing a bias at the start of the SVGD inference process. Similarly, we sample $\mathbf{\Theta}_{\mathcal{I},\text{init}} \sim \mathcal{N}(0, \sigma_{\mathcal{I},\text{init}}\mathbf{I})$ with $\sigma_{\mathcal{I},\text{init}}^2 = 0.1$.

For nonlinear Gaussians, we use the Glorot (sometimes called Xavier) normal to initialize the weights of the neural networks. [Glorot and Bengio, 2010].

## D.2 DATA GENERATION FOR THE SYNTHETIC CAUSAL INFERENCE TASKS

The data generation is done as follows: we first sample a random graph (either ER or SF) and then sample random parameters. We collect data samples by iterating through the topological ordering of the graph and sample a variable given its local parents. Since we consider additive Gaussian models, each variable is described by a distribution of the form $x_i|\mathbf{\Theta}, \mathbf{G} \sim \mathcal{N}(f(x_{\text{pa}_G(i)}, \mathbf{\Theta}), \sigma_i)$ where $f$ is either a linear function or a nonlinear feedforward neural network (cf. Sec. D.1). The noise variables $\sigma_i$ are sampled per variable and fixed once from $\sigma_i^2 \sim \mathcal{U}[0.05, 0.15]$. If the noise variables had the same variance across all variables in the graph, this would render identification possible [Peters and Bühlmann, 2014].

**Parameters.** The parameters and generative models are initialized as follows:

- *Linear BNs:* We sample the parameters $\mathbf{\Theta}$ uniformly and independently from $\mathcal{U}([-2, -0.5] \cup [0.5, 2])$ in order to bound the weights sufficiently away from zero.

- *Nonlinear BNs:* The NNs for each local conditional are the same model as used for BaCaDI as well as DCDI-G, that is, a fully connected NN with single hidden layer of size of 5 with biases. The nonlinear activation function is sigmoid. All weights and biases are drawn randomly and independently from a Gaussian $\mathcal{N}(0, 1)$.

**Interventions.** As described in the main text, we perform *hard* interventions on every node for the 20-node graphs. We create random values by first sampling $\hat{\mu}_{k,i} \sim \mathcal{N}(0,2)$ and then setting $\mu_{k,i}^I = \text{sign}(\hat{\mu}_{k,i}) \cdot 5 + \hat{\mu}_{k,i}$. This ensures that the interventions performed are bounded away from zero and out-of-distribution. If a variable $x_i$ is the target of the intervention in context $k$, we then have $p_i^{I_k}(x_i|\mathbf{\Theta}_{I_k}) = p_i^{I_k}(x_i|\mu_{k,i}^I) = \mathcal{N}(x_i; \mu_{k,i}^I, \sigma_I)$, where $\sigma_I^2 = 0.5$.

## D.3 DATA GENERATION WITH THE SERGIO GENE-EXPRESSION SIMULATOR

SERGIO [Dibaeinia and Sinha, 2020] is a single-cell expression simulator guided by Gene Regulatory Networks (GRN). The software tool can generate realistic single-cell transcriptomics datasets based on a user-defined graph input that describes the gene-regulatory network. SERGIO uses a stochastic differential equation (SDE), the so-called chemical Langevin equation (CLE), to simulate a gene's expression dynamics as a function of the changing levels of its regulators. Of interest to us, the simulations resemble the data collected by modern high-throughput, single-cell RNA sequencing (scRNA-seq) technologies. For more details, please consider the original paper by Dibaeinia and Sinha [2020]. We give a brief overview of how we simulate scRNA-seq data with SERGIO.

We use an implementation that is adapted from the open-surce code available under `https://github.com/PayamDiba/SERGIO` which is published under a GPL-3.0 license.

**Simulation.** SERGIO generates synthetic scRNA-seq data $\mathbf{D}$ for a given causal graph with $d$ genes in two stages: first, it collects clean gene expression snapshots; then they are altered by technical noise. The $N$ observations in $\mathbf{D}$ correspond to $N$ cell samples, i.e. one row in $\mathbf{D}$ describes the joint expression of the $d$ genes in a single cell.

In the first stage, SERGIO samples clean gene expressions through snapshots at random timesteps from the steady state of a dynamical system. In this regulatory process, the genes are expressed at rates influenced by other genes using the chemical Langevin equation (CLE). The source nodes in the causal graph $\mathbf{G}$ are denoted master regulators (MRs), whose expressions evolve at constant product and decay rates. The expressions of all downstream genes evolve non-linearly under production rates caused by the expression of their causal parents in $\mathbf{G}$. In addition, cell types are defined by specifications of the MR production rates, which significantly influence the evolution of the system. Thus, the data contains variation due to biological noise within collections of cells of the same type as well as across cell types. Ultimately, we generate single-cell samples collected from ten cell types [Dibaeinia and Sinha, 2020].

In the second stage, the clean gene expressions sampled previously are corrupted with technical noise that resembles the noise phenomena found in real scRNA-seq data. For simplicity, we do not make use of the technical noise for our experiments.

**Interventions.** We consider *knockout interventions* that clip a specific gene to zero. To that end, we extend SERGIO by forcing the production rate of knocked-out genes to be zero during simulation. As described in the main text, we do this for $M = 10$ gene targets and arrive at 11 contexts (including the observational setting).

**Parameters.** Given a causal graph $G$, the data generation to simulate $c$ cell types of $d$ genes is governed by the following parameters:

- $k \in \mathbb{R}^{d \times d}$ : interaction strengths (only used if edge $i \to j$ exists in $\mathbf{G}$)
    - We sample each $k_{i,j} \sim \mathcal{U}[1,5]$.
- $b \in \mathbb{R}_+^{d \times c}$ : MR production rates (only used if gene $j$ is a source node in $\mathbf{G}$)
    - We sample each $b_{i,j} \sim \mathcal{U}[1,3]$.
- $\gamma \in \mathbb{R}_+^{d \times d}$ : Hill function coefficients controlling nonlinearity of interactions
    - We fix a single $\gamma_{\text{const}} = 2.0$ and set $\gamma_{i,j} = \gamma_{\text{const}}$.
- $\lambda \in \mathbb{R}^d$ : decay rates per gene
    - We fix each $\lambda_i = 0.8$.
- $\zeta \in \mathbb{R}_+^d$ : scale of stochastic process noise in chemical Langevin equation simulating the dynamical system.
    - We fix $\zeta = 1.0$.

We standardize the collected data for all methods by subtracting the empirical mean and dividing by the standard deviation.

## D.4 BASELINES & IMPLEMENTATION

In this section, we provide additional details on the baseline methods and their implementations used in our experiments. Our source-code and instructions on how to reproduce results are available under `https://www.dropbox.com/sh/5vtj4zp9h8sr9zq/AACkJ5naAxhXnpIByNEezI4Ia?dl=0`.

For all methods, we use implementations adapted from the source code of DCDI available under `https://github.com/slachapelle/dcdi` (where the authors also included their baselines' code). The implementation of JCI-PC is a modified version of the R package using code from the JCI repository `https://github.com/caus-am/jci/tree/master/jci`. UT-IGSP has an implementation available from `https://github.com/uhlerlab/causaldag`.

**DAG Bootstrap.** To compare all methods for Bayesian model averaging, we employ the nonparametric DAG bootstrap [Friedman et al., 2013]. It performs model averaging by bootstrapping the observations $\mathbf{D}$ to yield a collection of synthetic data sets, each of which is used to learn a single graph. We sample with replacement for each dataset $\mathcal{D}_k$ individually. The collection of unique single graphs approximates the posterior by weighting each graph by its unnormalized posterior probability.

All of the methods are evaluated with 20 bootstrap samples, i.e. the same number of samples as particles used for the SVGD instantiation of BaCaDI. The only exception is DCDI-G, where we only use 5 bootstrap samples due to longer runtimes.

**JCI-PC.** The Joint Causal Inference (JCI) framework [Mooij et al., 2016] introduces a general formulation to extend the initial causal graph by auxiliary nodes that describe the different contexts, effectively performing causal discovery over a graph of size $d + M$. This can be instiantiated with different standard algorithms. We use the PC algorithm [Spirtes et al., 2000] that relies on conditional independence tests (CI) to discover the Markov Equivalence Class (MEC), i.e. the skeleton of a graph with v-structures as well as possible identifiable edge directions. We use the Gaussian CI tests that are best-suited for the Gaussian BNs that we consider with the threshold parameter $\alpha^{\text{JCI}}$.

For computing the graph metrics, we compute the $\mathbb{E}$-SID between the CPDAGs of the GT graph and predictions of JCI-PC. Additionally, we favor JCI-PC when computing AUPRC scores. See Sec. D.5 for more details.

To arrive at a DAG that we can use to estimate MLE parameters and compute log-likelihood metrics, we generate a consistent expansion of the CPDAG as defined by Chickering [2002]. That is, we generate a DAG s.t. the CPDAG has the same skeleton and v-structures and every directed edge in the CPDAG has the same direction in the DAG. To that end, we perform a random consistent expansion as described by Dor and Tarsi [1992].

**UT-IGSP.** The *interventional greedy sparsest permutation* (IGSP) method [Wang et al., 2017, Yang et al., 2018] proposes an algorithm that learns causal structures via local scores based on CI relations and permutation search. The work is extended by Squires et al. [2020] to the case of unknown targets (UT-IGSP). Analogous to JCI-PC, we make use of Gaussian CI and invariance tests with parameters $\alpha^{\text{UT-IGSP}}$ and $\alpha^{\text{UT-IGSP}}_{\text{inv}}$. Since we consider a low sample setting, it is possible that the algorithm as provided by their open source implementation does not succeed when computing the correlation matrix. This is because it computes a correlation matrix which becomes singular in case the number of samples is smaller than the number of variables considered, thus rendering an inversion of this matrix impossible. In case this happens, we retry the inference with a halved $\alpha^{\text{UT-IGSP}}$ confidence threshold. The maximum number of restarts is set to 10. Should this maximum be reached, the bootstrap sample is simply dropped.

Similar to JCI-PC, for computing the graph metrics, the $\mathbb{E}$-SID is computed as the midpoint of the lower and upper bound between the CPDAGs of the GT graph and predictions of UT-IGSP. Additionally, we favor UT-IGSP when computing AUPRC scores. See Sec. D.5 for more details.

Moreover, we obtain a DAG by the random consistent expansion of the CPDAG (as described above for JCI-PC) to compute the log-likelihood metrics.

**DCDI-G.** The work of Brouillard et al. [2020] introduced a model for *Differentiable Causal Discovery from Interventional Data* (DCDI) that performs causal structure learning via the augmented Lagrangian method. They do so by formulating a continuous-constrained optimization problem that relies on stochastic gradient descent and neural networks to fit the local conditionals.

For a fair comparison, we employ the exact same model as used for the nonlinear Gaussian BNs used in BaCaDI, that is, a feedforward neural network with one hidden layer of size 5 and Gaussian additive noise. This model is also called DCDI-G in [Brouillard et al., 2020], however, a smaller model than the default configuration that was used in their paper (2 hidden layers with 16 hidden units); with that, we performed initial experiments, but saw strong overfitting and thus reduced

model-capacity. As elementwise activation function, we use the leaky ReLU with negative slope of $0.25$ as suggested by their work. Additionally, to have comparable log-likelihood metrics, we fix the noise variables to $\sigma^2 = 0.1$ just as done for BaCaDI. These noise variables could generally be learned by both models.

Since DCDI-G takes longer to compute, we restrict the number of bootstrap samples to 5.

## D.5 METRICS

Here, we describe the evaluation metrics more in detail.

Our reported metrics focus on three essential aspects of our inference problem: causal graph prediction, intervention detection, and learning the local conditionals of individual variables/nodes. We describe the metrics in the following.

- *SID:* The *Structural Interventional Distance* (SID) [Peters and Bühlmann, 2015] quantifies the closeness between two DAGs in terms of how well their interventional adjustment sets coincide. Since we perform posterior inference and arrive at a distribution over graphs, we consider the *expected* SID:

$$\mathbb{E}\text{-SID}(p, \mathbf{G}_{\text{gt}}) := \sum_{\mathbf{G}} p(\mathbf{G}|\mathcal{D}) \cdot \text{SID}(\mathbf{G}, \mathbf{G}_{\text{gt}}) \tag{36}$$

  We use the implementation provided by the Causal Discovery Toolbox [Kalainathan and Goudet, 2019].

  Since UT-IGSP and JCI-PC only return a CPDAGs of the Interventional Markov Equivalence Class (I-MEC), we calculate its lower and upper bound of SIDs in the I-MEC, and report their midpoint as the $\mathbb{E}$-SID. Note that the DAG bootstrap variants for all baselines as well as BaCaDI use the weighted mixture rather than the empirical distribution of samples; the weight is based on the achieved unnormalized log-likelihood on the bootstrap sample.

- *SHD*: Another commonly used metric is the structural hamming distance (SHD) that reflects the graph edit distance to the ground truth. However, it often does not properly reflect the closeness of two DAGs in terms of their causal interpretation. For instance, the trivial prediction of the empty graph achieves competitive SHD scores for the sparse graphs we consider in this paper. In our main text, we thus focus on the $\mathbb{E}$-SID as well as other metrics that together better assess the quality of causal graph predictions. For completeness, we report the $\mathbb{E}$-SHD results in Appx. E.6.

- *Threshold metrics*: Treating the edge prediction as a classification task, we compute the area under the precision recall curve (AUPRC) for pairwise edge prediction based on the posterior marginals $p(g_{ij} = 1|\mathbf{D})$. This marginal is simply defined as averaging the presence of edges in the posterior samples: $p(g_{ij} = 1|\mathbf{D}) = \mathbb{E}_{p(\mathbf{G}|\mathbf{D})}\mathbf{1}[g_{ij} = 1]$. Note that this is a more suitable metric than, e.g., the AUROC, as sparse graphs translate to highly imbalanced edge classification tasks.

  For both baselines JCI-PC and UT-IGSP, which both possibly return a CPDAG with edges that are undecided, we favor them when computing the AUPRC metric. We orient a predicted undirected edge correctly when a ground truth edge exists and only count a falsely predicted undirected edge as a single mistake.

- *Interventional AUPRC*: Similarly, we report the *interventional* AUPRC (INTV-AUPRC) for the classification of targets. This again captures how well an algorithm predicts which variables have been intervened on. Since we are performing sparse interventions, this better captures a model's performance for a highly imbalanced classification task.

- *I-NLL:* We compute the average negative *interventional log-likelihood (I-LL)* on $M^{\text{test}} = 10$ heldout interventional datasets $\mathbf{D}^{\text{test}} = \{\mathcal{D}_1^{\text{test}}, ..., \mathcal{D}_{10}^{\text{test}}\}$, where different interventions are performed compared to the training datasets. Each interventional test dataset comprises 100 samples, and has known intervention targets $I_{\text{test},k}^{\text{tar}}$ and effect distributions $p(x_i|\mathbf{\Theta}_{I_{\text{test},k}})$
  The I-NLL is computed via

$$\text{I-NLL}(p, \mathbf{D}^{\text{test}}) := -\frac{1}{M^{\text{test}}} \sum_{k=1}^{M^{\text{test}}} \mathbb{E}_{p(\mathbf{G}, \mathbf{\Theta}|\mathbf{D})} \left[ \frac{1}{|\mathcal{D}_k^{\text{test}}|} \log p(\mathcal{D}_k^{\text{test}}|\mathbf{G}, \mathbf{\Theta}, I_{\text{test},k}^{\text{tar}}, \mathbf{\Theta}_{I_{\text{test},k}}) \right] .$$

  Since UT-IGSP and JCI-PC are not equipped with local conditional distributions, we use the linear Gaussian maximum-likelihood parameters (MLE) that can be computed in closed-form [Hauser and Bühlmann, 2014] to compute the heldout I-NLL. Note that for the nonlinear datasets, this creates a model mismatch since the closed form is only available for the linear Gaussian BNs.

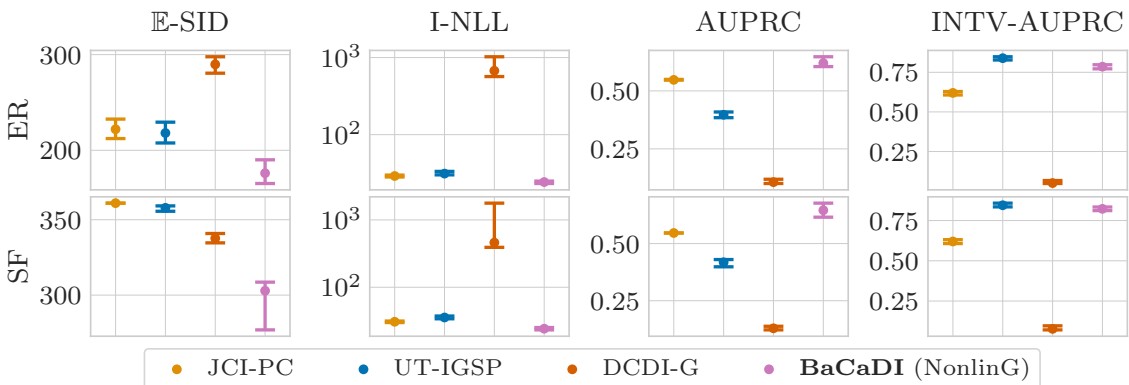

Figure 6: **Nonlinear Gaussian.** Joint posterior inference over BCNs and interventions for *nonlinear Gaussian* ground-truth BCNs. The results are for data from ER-2 (top) and SF-2 (bottom) graphs with $d = 20$ variables and $M = 20$ contexts. BaCaDI consistently gives the best causal structure and intervention predictions.

## D.6 HYPERPARAMETER SEARCH

To ensure a fair comparison, we perform a hyperparameter search for all baselines. The search ranges can be found in Table 1. Throughout all experiments, we use at least 20 hyperparameter samples and aggregate results over 30 different random seeds and graphs. For the results of BaCaDI in the main text, we fix the hyperparameters $\alpha = 0.01$, $\beta = 2$ and $\lambda = 1$, which are the linear scale for the temperature parameter of the sigmoids, the linear scale for the sparsity regularizer and the sparse intervention regularizer, respectively.

| Method | Hyperparameter | Comment | Search range |
|---|---|---|---|
| BaCaDI | $\tau_Z$ | Kernel lengthscale | $\log_{10} \mathcal{U}[-1, 1.7]$ |
| | $\tau_\gamma$ | Kernel lengthscale | $\log_{10} \mathcal{U}[-1, 1.7]$ |
| | $\tau_\theta$ | Kernel lengthscale | $\log_{10} \mathcal{U}[1.2, 5]$ |
| JCI-PC | $\alpha^{\text{JCI}}$ | CI tests | $\log_{10} \mathcal{U}[-5, -1]$ |
| UT-IGSP | $\alpha^{\text{UT-IGSP}}$ | CI tests | $\log_{10} \mathcal{U}[-5, -1]$ |
| | $\alpha_{\text{inv}}^{\text{UT-IGSP}}$ | Invariance tests | $\log_{10} \mathcal{U}[-5, -1]$ |
| DCDI-G | batch size | - | $\mathcal{U}(\{16, 32, 64\})$ |
| | $\lambda_R$ | Sparsity coefficient for interventions | $\log_{10} \mathcal{U}[-8, -1]$ |
| | $\lambda$ | Sparsity coefficient for graph | $\log_{10} \mathcal{U}[-3, -1]$ |
| | $h$ | Convergence threshold | $\log_{10} \mathcal{U}[-8, -6]$ |

Table 1: Hyperparameter search space for all methods

## E ADDITIONAL EXPERIMENTS

### E.1 SYNTHETIC NONLINEAR DATASETS

In addition to the results for synthetic linear Gaussian BNs in Sec. 6, we perform experiments for synthetic *nonlinear* Gaussian BNs. As described in Sec. D.1 and Sec. D.2, the local conditionals are modelled by feedforward neural networks and hence the same model as used DCDI-G and the nonlinear BaCaDI. The corresponding results can be found in Fig. 6.

Analogous to the previous evaluations, BaCaDI gives predictions that are the closest to the ground-truth CBN. It strongly outperforms the baselines in terms of the $\mathbb{E}$-SID and AUPRC while achieving low I-NLL and high INTV-AUPRC. Among the baselines, UT-IGSP is the best at detecting interventions, largely on par with BaCaDI, but it fails behind in the other metrics.

## E.2 LARGER DATASETS

We report additional results when doubling the dataset size, i.e. we collect $n_0 = 200$ observational samples and $n_k = 20$ samples per interventional context. The results for synthetic linear and nonlinear Gaussian BNs can be found in Fig. 7 and Fig. 8, respectively.

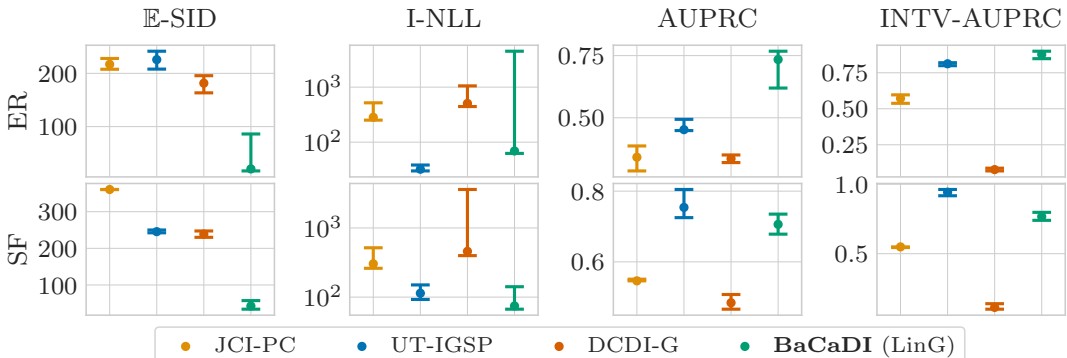

Figure 7: **Linear Gaussian with more data.** Joint posterior inference over BCNs and interventions for *linear Gaussian* ground-truth BCNs. The results are for data from ER-2 (top) and SF-2 (bottom) graphs with $d = 20$ variables and $M = 20$ contexts. We here double the dataset size to $N = 600$ samples in total. Similar to the previous setting with lower sample sizes, BaCaDI consistently is the most competitive method.

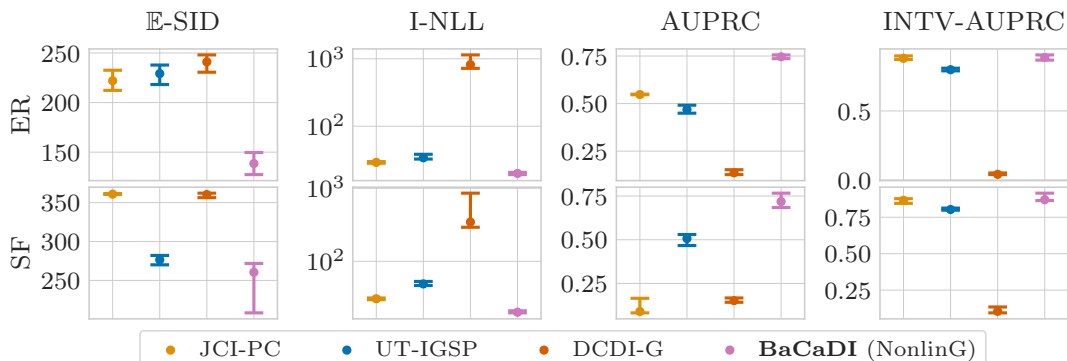

Figure 8: **Nonlinear Gaussian with more data.** Joint posterior inference over BCNs and interventions for *nonlinear Gaussian* ground-truth BCNs. The results are for data from ER-2 (top) and SF-2 (bottom) graphs with $d = 20$ variables and $M = 20$ contexts. We here double the dataset size to $N = 600$ samples in total. Again, BaCaDI performs the best across all metrics.

## E.3 CALIBRATION

Having compared BaCaDI with the baselines across the metrics for causal structure learning in the previous sections and the main text, we now provide additional experiments to show how BaCaDI incorporates *epistemic uncertainty*. To quantify the reliability of our uncertainty estimates, we utilize the concept of *calibration* [Gneiting et al., 2007, Kuleshov et al., 2018] to quantify the reliability of our uncertainty estimates. We show the results in Appx. E.3. Notably, BaCaDI is the only method that takes into account the *epistemic uncertainty* when dealing with limited data and shows that its probabilistic predictions are reliable. Formally, we consider a probabilistic predictor to be well calibrated if in expectation, its $\alpha$-% confidence intervals cover $\alpha$-% of the true targets.

We visualize this in Fig. 9 for the task of predicting edges in CBNs. The plot compares the confidence intervals for varying thresholds of $\alpha$ with the empirical coverage, that is, the average percentage of edges present in the ground-truth graphs. Perfectly calibrated predictions exhibit a proportional one-to-one relationship between confidence level and empirical coverage, indicated by the black straight line in Fig. 9. All three baseline methods are severely underconfident in their predictions. BaCaDI, in contrast, produces edge predictions that are close to the ideal calibration line. This indicates that

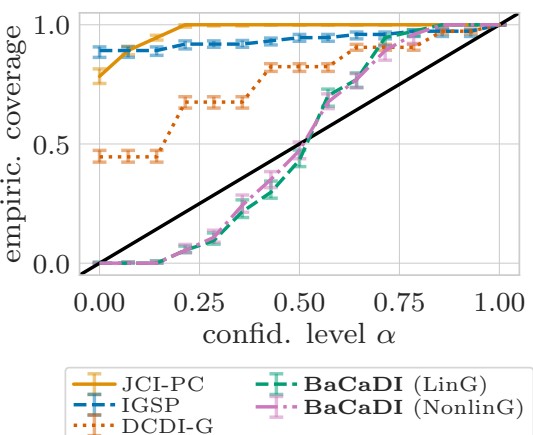

Figure 9: **Calibration plot** for edge predictions with SERGIO simulated data. The x-axis shows varying thresholds of the confidence level $\alpha$ compared to the empirical coverage of the edges on the y-axis. Error bars show the 90% confidence intervals across datasets. Only BaCaDI makes well-calibrated predictions.

BaCaDI properly takes into account the *epistemic uncertainty* when dealing with limited data and shows that its probabilistic predictions are reliable. Additionally, BaCaDI offers the possibility to fine-tune the calibration of the predictor through the SVGD kernel parameters.

**Calibration for synthetic datasets.** We show the calibration for all methods with the synthetic datasets used in the main text. The calibration plots can be found in Fig. 10. Notably, BaCaDI is the only method that does not give grossly overconfident predictions. Instead, it is conservative in predicting edges for the linear BNs, and gets closest to the ideal calibration line for nonlinear BNs.

**Ablation study.** In addition to well capturing the causal mechanisms when performing posterior inference, we now demonstrate the flexibility of calibrating the inference of BaCaDI. Through the use of SVGD and a kernel that can be designed specifically for the application, BaCaDI can be fine-tuned to a desired calibration. We show different inference results in Fig. 11 for linear as well as nonlinear datasets. Depending on the lengthscale hyperparameters for the kernel in 28, we see that the predictions of BaCaDI can be easily adapted to varying levels. It thus properly takes into account the epistemic uncertainty when dealing with limited data.

### E.4    OBSERVATIONAL VS. INTERVENTIONAL DATA

As an interesting use case, we now evaluate how interventional data helps predicting the causal structure with BaCaDI. In general, intervening on variables in the system and observing the outcome provides information that helps discovering the causal mechanisms; in the perfect setting, when the interventions are fully known, they increase identifiability (reducing the interventional Markov equivalence class) [Hauser and Bühlmann, 2012]. However, the information gain can be limited when the interventions are unknown [Squires et al., 2020].

We investigate how BaCaDI can leverage such unknown interventions compared to using just observational data. To that end, we evaluate 3 different settings: i) when only observational data is available, ii) interventional data with full knowledge of the interventions, and iii) unknown interventions. When the interventions are known, the posterior inference of BaCaDI can be reduced to Eq. 1.

We consider nonlinear Gaussian BNs for $d = 20$ node graphs. All methods receive the exact same number of samples for inference. That is, we collect $n_0 = 300$ samples from the ground-truth BN without interventions for the observational dataset; the interventional datasets, as before, have $n_0 = 100$ observations and $n_k = 10$ samples per interventional context. We report the results in Fig. 12.

We can see how BaCaDI achieves much better results when interventional data is available. The case of known interventions serves as a natural baseline, where BaCaDI achieves the best performance and gets closest in recovering the true causal structure. This is most clearly shown by the $\mathbb{E}$-SID and AUPRC scores. Notably, BaCaDI is able to get close this baseline even when interventions are unknown and outperforms the setting where just observational data is available. This shows the

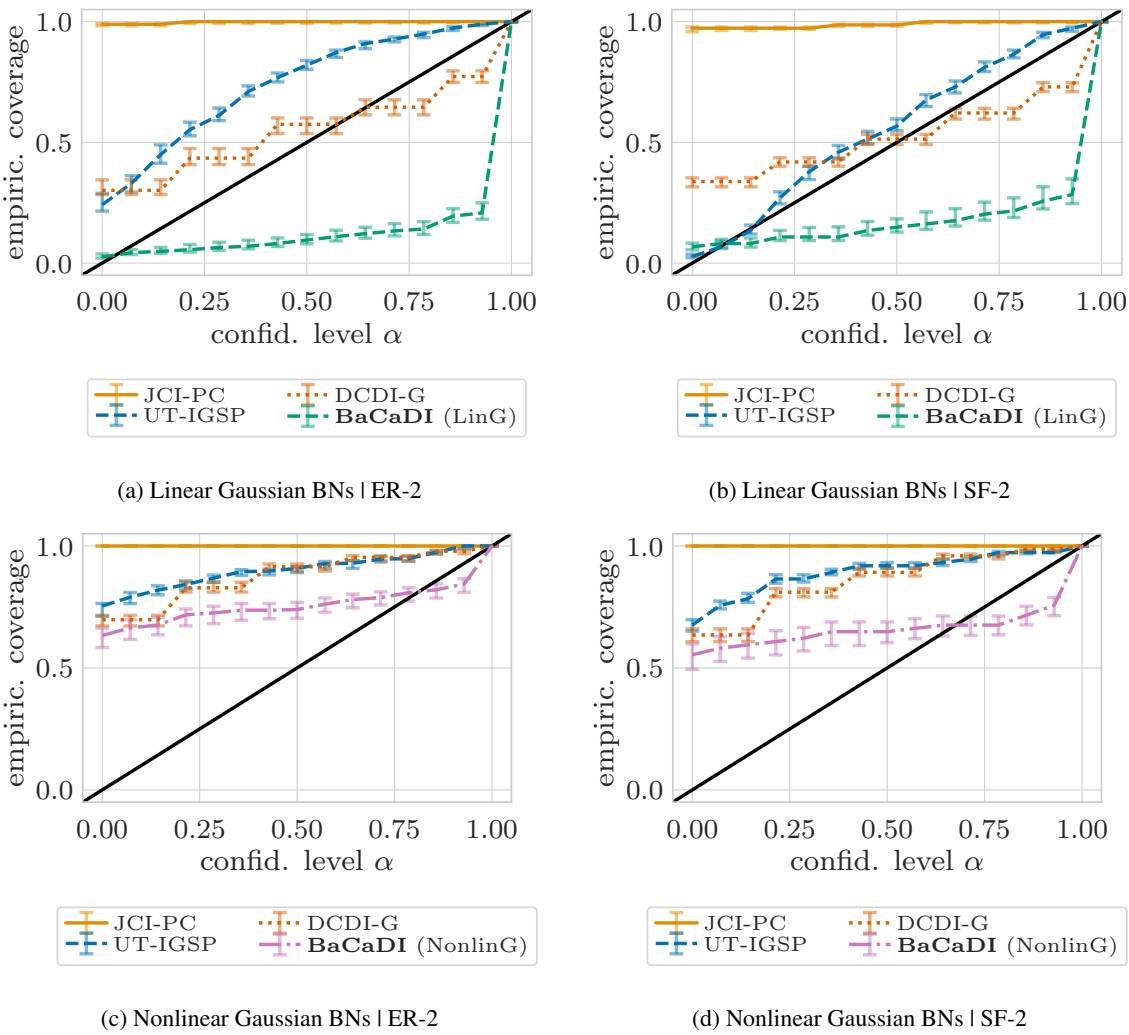

Figure 10: **Calibration plots for the synthetic datasets.** The top row shows the calibration plots for linear Gaussian BNs for ER graphs (left) and SF graphs (right). The bottom row shows the nonlinear Gaussian BNs. All results correspond to the same models and configuration as the results reported in the main text. Notably, BaCaDI is the closest to the ideal calibration line for nonlinear BNs.

benefit of performing and collecting interventions even when some of the effects may be unknown and is a promising result for future work.

### E.5  50 NODE GRAPHS

As additional experiments, we perform posterior inference for larger graphs with 50 nodes in total. Analogous to the previous evaluations, we consider synthetic linear Gaussian BNs with hard interventions on every node. We use a larger dataset of $n_0 = 200$ observational samples and $n_k = 20$ samples per interventional context. We show the results in Fig. 13. We see how BaCaDI scales to larger graphs, performing competitively across all metrics. In particular, it outperforms the baselines by a large margin in terms of the $\mathbb{E}$-SID and AUPRC for ER graphs.

### E.6  STRUCTURAL HAMMING DISTANCE

For completeness, we include the *Expected Structural Hamming Distance* ($\mathbb{E}$-SHD) metric for all results discussed in the main text for $d = 20$ node graphs in Table 2. Analogous to the $\mathbb{E}$-SID, the $\mathbb{E}$-SHD is defined as $\mathbb{E}\text{-SHD}(p, \mathbf{G}_{\text{gt}}) :=$

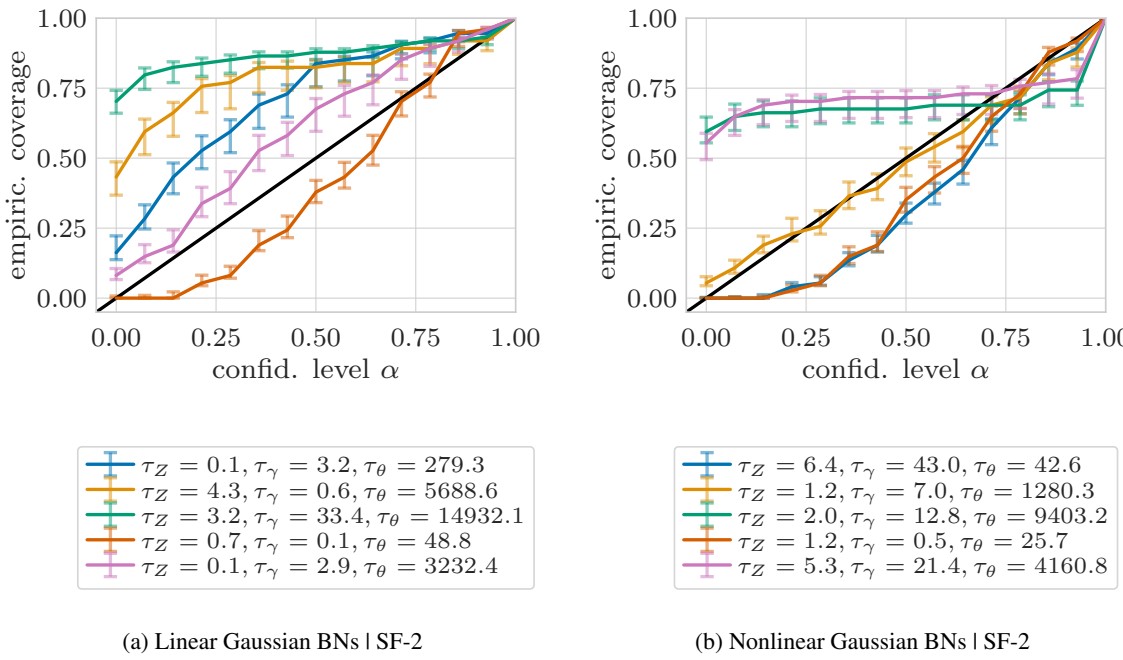

(a) Linear Gaussian BNs | SF-2                                  (b) Nonlinear Gaussian BNs | SF-2

Figure 11: **Calibration plot sweeps**. We show calibration plots for varying kernel lengthscales for $d = 20$ node SF graphs for linear Gaussian BNs (left) and nonlinear Gaussian BNs (right). BaCaDI offers the possibility to fine-tune the calibration of the predictive model depending on the application and desired results. It thus can properly take into account the *epistemic uncertainty* when dealing with limited data.

$\sum_{\mathbf{G}} p(\mathbf{G}|\mathcal{D}) \cdot \text{SHD}(\mathbf{G}, \mathbf{G}_{\text{gt}})$. The SHD simply reflects the graph edit distance where wrongly inverted edges are counted as only one error. However, while it captures closeness to the ground truth graph, trivial predictions like the empty graph achieve competitive results for sparse graphs. For example, in the setting of $d = 20$ nodes and $2d$ edges in expectation, the empty graph will achieve a $\mathbb{E}$-SHD of $40$ in expectation. Similar conclusions hold if only a handful of edges are predicted. In the main text, we thus resort to the $\mathbb{E}$-SID as well as additional metrics that together establish the quality of the causal predictions.

| | JCI-PC | UT-IGSP | DCDI-G | BaCaDI (LinG) | BaCaDI (NonlinG) |
|---|---|---|---|---|---|
| **Linear Gaussian BNs graphs:** | | | | | |
| ER | 38.97 (1.65) | 41.38 (2.53) | 63.29 (4.52) | 50.32 (10.98) | - |
| SF-2 | 37.42 (0.32) | 34.35 (1.18) | 51.68 (3.67) | 36.48 (6.12) | - |
| **Nonlinear Gaussian BNs graphs:** | | | | | |
| ER | 38.93 (1.65) | 35.09 (1.76) | 64.91 (2.61) | - | 18.73 (1.85) |
| SF-2 | 37.00 (0.00) | 33.75 (0.48) | 57.92 (2.69) | - | 23.55 (1.43) |
| **SERGIO graphs** | | | | | |
| SF-2 | 44.29 (0.70) | 45.05 (1.88) | 56.39 (1.63) | 108.71 (3.75) | 115.04 (6.22) |

Table 2: **Expected Structural Hamming Distance.** We report the $\mathbb{E}$-SHD for all methods for the main results with $d = 20$ node graphs. We aggregate results over 30 different seeds and report the mean and standard error. While the $\mathbb{E}$-SHD is a simple and commonly used metric, it can be misleading. For sparse graphs, the trivial prediction of the empty graph achieves competitive $\mathbb{E}$-SHD scores; similar assessments can be made for predictions with only a few edges. We include the results for completeness.

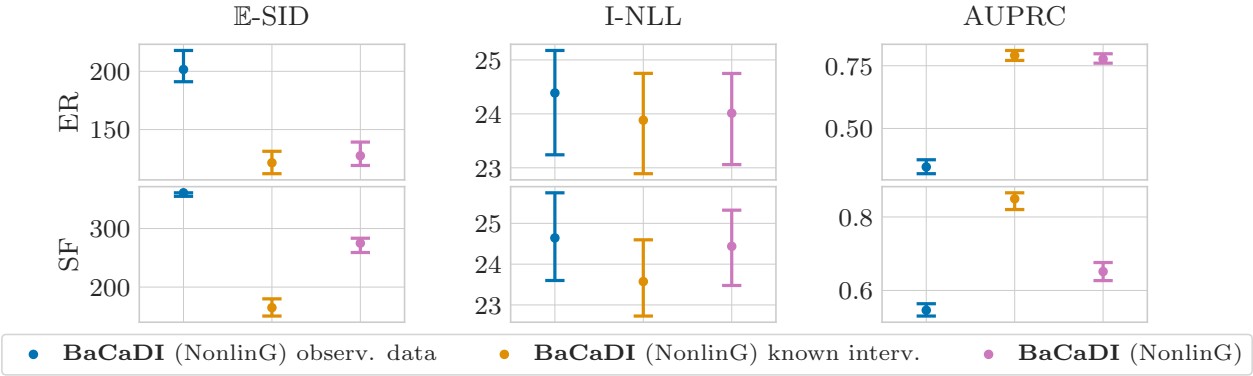

Figure 12: **Comparison: Observational data vs. known and unknown interventions.** Additional results comparing BaCaDI with observational data against known and unknown interventions for nonlinear Gaussian BNs for $d = 20$ node graphs. As a natural baseline, the setting of knowing the intervention targets and effects leads to the best performance across all metrics. When the interventions are unknown, BaCaDI achieves results close to this baseline. Notably, it outperforms the setting where just observational data is available.

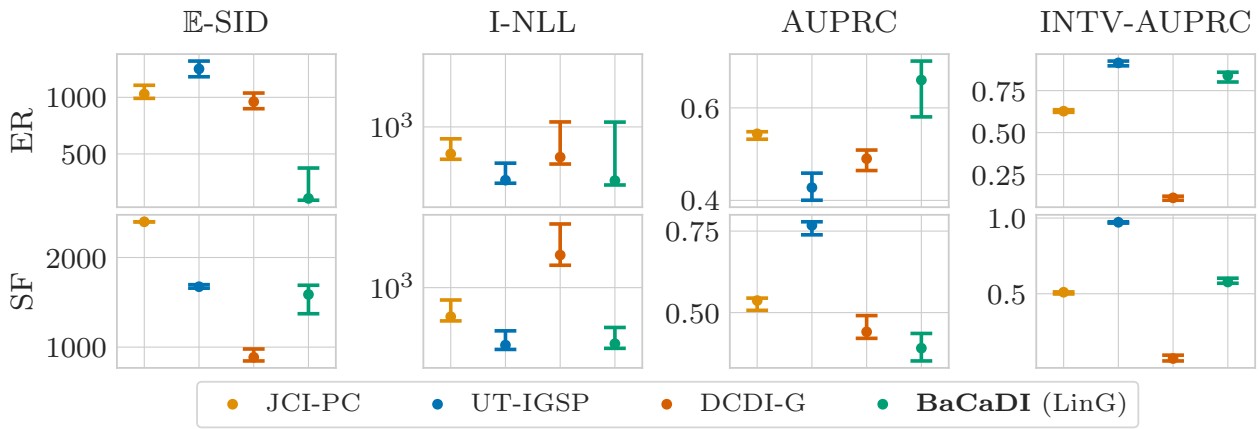

Figure 13: **Linear Gaussian BNs with 50 nodes.** Additional results comparing BaCaDI on larger graphs with $d = 50$ nodes for Linear Gaussian BNs for ER-2 (top) and SF-2 (bottom). BaCaDI performs competitively across all metrics. In particular, it makes significantly better causal mechanism predictions for ER graph as captured by the $\mathbb{E}$-SID.

### E.7   RUNTIMES

We report the average runtimes (in minutes) for all methods and main results for $d = 20$ node graphs in Table 3.

### E.8   COMPUTATIONAL RESOURCES

All experiments reported in this paper were performed in bulk and in parallel on 2-core CPU nodes of an internal cluster. No GPUs were used. We estimate the total compute to roughly amount to 40000 hours of CPU time.

|       | JCI-PC | UT-IGSP | DCDI-G | BaCaDI (LinG) | BaCaDI (NonlinG) |
|-------|--------|---------|--------|---------------|------------------|
| **Linear Gaussian BNs graphs:** | | | | | |
| ER | 2.07± 0.16 | 1.31± 0.22 | 260.03±56.11 | 51.68±13.12 | - |
| SF-2 | 2.17± 0.15 | 1.46± 0.09 | 176.89±30.46 | 60.57±22.95 | - |
| **Nonlinear Gaussian BNs graphs:** | | | | | |
| ER | 1.89± 0.07 | 1.16± 0.06 | 143.88±12.53 | - | 491.26±91.74 |
| SF-2 | 1.92± 0.06 | 1.41± 0.06 | 145.43±25.83 | - | 519.79±97.95 |
| **SERGIO graphs** | | | | | |
| SF-2 | 2.36± 0.09 | 0.42± 0.20 | 128.04± 9.97 | 45.11± 1.52 | 323.11±101.15 |

Table 3: **Average runtimes** for all methods for the main results with $d = 20$ node graphs. We aggregate results over 30 different seeds and report the mean and standard deviation. All numbers given are in minutes.

# F STEIN VARIATIONAL GRADIENT DESCENT

We here give the most important points of the Stein Variational Gradient Descent (SVGD) introduced by Liu and Wang [2016]. Fundamentally, the work of Liu and Wang [2016] connects the mathematical notions of probability discrepancies with a variational inference method, which closely resembles the gradient descent algorithm. For a comprehensive overview, we refer to the original paper.

**Stein's identity and discrepancy.** Formally, let $p(\mathbf{x})$ be a continuously differentiable (i.e. smooth) density supported on $\mathcal{X} \subseteq \mathbb{R}^d$. For a smooth vector function $\phi(\mathbf{x})$, the *Stein's identity* states that for sufficiently regular $\phi$, we have

$$\mathbb{E}_{\mathbf{x} \sim p}[\mathcal{A}_p \phi(\mathbf{x})] = 0, \tag{37}$$

$$\text{where } \mathcal{A}_p \phi(\mathbf{x}) = \phi(\mathbf{x}) \nabla_x \log p(\mathbf{x})^T + \nabla_x \phi(\mathbf{x}) \tag{38}$$

where $\mathcal{A}_p$ is the Stein operator acting on the function $\phi$. This equation can be subsequently used as a discrepancy measure: when considering the expectation over a smooth density $q$ different than $p$, we obtain the so called *Stein discrepancy* by considering the "maximum violation of Stein's identity". That is, we have

$$\mathbb{S}(q,p) = \max_{\phi \in \mathcal{F}}[\mathbb{E}_{\mathbf{x} \sim q}[\text{trace}(\mathcal{A}_p \phi(\mathbf{x}))]^2] \tag{39}$$

for a choice of a set of functions $\mathcal{F}$, for instance the reproducing kernel Hilbert space (RKHS) denoted by $\mathcal{H}^d$.

**Variational Inference using smooth transforms.** The goal of variational inference is to approximate a target distribution $p$ using a simpler distribution $q^*$ which minimizes the KL-divergence

$$q^* = \arg\min_{q \in \mathcal{Q}} KL(q||p) \tag{40}$$

In order to minimize the KL-divergence, the authors in [Liu and Wang, 2016] consider incremental transforms formed by a small perturbation to the identity map that make up the set of distributions $\mathcal{Q}$. That is, the transform $\mathbf{T} : \mathcal{X} \to \mathcal{X}$ is defined as

$$\mathbf{T}(\mathbf{x}) = \mathbf{x} + \epsilon\phi(\mathbf{x}) \tag{41}$$

where $\phi$ is a smooth function that characterizes the perturbation direction. As a key result, Liu and Wang [2016] connect these transforms to the Stein operator and the derivative of the KL divergence. The authors show that if the function $\phi$ lies in the ball of the vector valued RKHS $\mathcal{H}^d$, the direction of the steepest descent on the KL divergence between a fixed $q$ and the target $p$ is given by

$$\phi_{q,p}^* = \mathbb{E}_{\mathbf{x} \sim q}[k(\mathbf{x}, \cdot)\nabla_{\mathbf{x}} \log p(\mathbf{x}) + \nabla_{\mathbf{x}} k(\mathbf{x}, \cdot)] \tag{42}$$

where one can easily identify the form of the Stein operator. What is more, the value of the obtained gradient equals the (negative) kernelized Stein discrepancy measure $-\mathbb{S}(q,p)$.

**General Algorithm.** This mathematical result suggests an iterative method that transforms an initial reference distribution $q_0$ to the target distribution $p$. Starting with a finite set of random particles $\{\mathbf{x}^{(m)}\}_{m=1}^M$, for some iteration $t$ each particle is updated deterministically according to

$$\mathbf{x}_{t+1}^{(m)} \leftarrow \mathbf{x}_t^{(m)} + \epsilon_t \phi(\mathbf{x}_{t+1}^{(m)}) \tag{43}$$
where

$$\phi(\mathbf{x}) = \frac{1}{M}\sum_{l=1}^M k(\mathbf{x}_{t+1}^{(l)}, \mathbf{x})\nabla_{\mathbf{x}} \log p(\mathbf{x}) + \nabla_{\mathbf{x}} k(\mathbf{x}_{t+1}^{(l)}, \mathbf{x}) \tag{44}$$

These steps iteratively decrease the KL divergence between $q_t$ and $p$, ultimately converging.

Importantly, the advantage of SVGD is that it only depends on the gradient of the kernel $k(x, \cdot)$ that can be defined for the application, as well as the score function $\nabla_{\mathbf{x}} \log p(x)$ which can be computed without knowing the (intractable) normalization constant of $p$. On an intuitive level, the first part of the perturbation direction expression drives the particles to high density regions close to the mode of the target distribution $p$, whereas the term $\nabla_{\mathbf{x}} k(\mathbf{x}, \cdot)$ acts as a repulsive force that prevents the particles from collapsing together.