# OpenReview forum: "BaCaDI: Bayesian Causal Discovery with Unknown Interventions"
_auai.org/UAI/2022/Workshop/CRL — CRL@UAI 2022 Poster_

### Official Review · Reviewer_Ah4Z · 2022-06-29
**comments for the authors**

**Rating:** 9
**Confidence:** 4

**Review:**

Summary: This paper is about inferring causal graphs with samples from multiple interventional distributions (and the observational distribution), but where the targets of interventions are unknown. The proposed method also infers the unknown intervention targets. The method is motivated by examples in biology where we may know that gene knockout experiments were done but we don't know all the effects of the knockouts. The paper addresses several technical challenges in this setting. First, they posit priors on the structure of intervention targets so that we can encode domain knowledge like the idea that intervention targets are sparse. Second, they formulate expressions for the posterior over intervention targets and graphs that lead to a differentiable objective over continuous parameters instead of discrete objects. Finally, they present an algorithm based on variational inference for approximating the posterior and inferring causal graphs and intervention targets. The paper presents empirical studies on different types of simulated and compares several methods that are applicable in the causal discovery with unknown intervention setting. The studies suggest that the proposed method outperforms related approaches in these tasks for a range of evaluations.

Overall, I think this is an extremely well-written, clear and strong contribution. The problem was well motivated and explained, discussing related work. The background and added challenges of the unknown interventions setting were also clearly motivated, and the technical details of the method were also clear (though much of the detail is in the appendix). I think that there are lots of exciting questions to explore in proposing interesting priors for intervention targets that could also take into account structure across intervention settings. For example, perhaps some key genes are never the targets of interventions because if they were, the cell would die. Or, we might want to model the correlation structure among the intervention targets more directly.

I'm very enthusiastic about this paper in general. My main feedback to the authors is to ask for more details about the limitations of the framework and the (un)identifiability of intervention targets in the model. What can we say about the equivalence class of intervention targets that all lead to the same marginal likelihood of the observations? Can we say anything about this particular model and the identifiability of the parameters that govern causal effects, graphs, interventions, etc.

---

### Official Review · Reviewer_D39g · 2022-07-03
**Review of "BaCaDI: Bayesian Causal Discovery with Unknown Interventions"**

**Rating:** 5
**Confidence:** 5

**Review:**

The authors introduce a differentiable, score-based, Bayesian method for causal structure learning in the presence of interventions with possibly unknown targets. They provide experiments showing that the method outperforms existing methods on both synthetic data and semi-synthetic data in a gene regulation application.

The paper fills an obvious gap in the literature, combining a recently-introduced variational method for causal structure learning (Lorch et al., 2021) with existing techniques for converting causal structure learning algorithms from the observational setting to the unknown-target interventional settings (Mooij et al., 2016, Squires et al., 2020) by introducing binary variables to represent interventions. This composes a solid contribution for a workshop paper, but the topic is at least somewhat outside of the intended scope of the workshop - no causal representation learning is occurring, rather, the authors consider the standard setting in which the causal variables are directly observed.

**Pros:**
- The paper introduces a promising method for a relevant task in causal structure learning.
- The paper is written with decent clarity.
**Cons:**
- The paper is not about “causal representation learning” proper.
- For a workshop paper, the paper is fairly technical with a lengthy appendix. It may be more useful as a conference paper, especially if this leads to additional methodological innovation. The paper as it exists is more about bringing together existing ideas, so that it isn’t very original.
**Minor suggestions:**
- Make it clear in the reference to Figure 4 on page 4 that the figure is in the appendix.
- In the interventional prior term in Equation (2), should the conditional distribution of $\Theta_{I_k}$ include conditioning on $\Theta$? Otherwise I do not see how the consistency between interventional and observational parameters is enforced. This question applies in other spots such as Equation (3).

---

### Meta-Review · Program_Chairs · 2022-07-06

**Recommendation:** Accept (Poster)
**Confidence:** 4

**Metareview:**

While more focused on causal discovery, the bayesian framework with unknown intervention will likely be an interesting addition to the workshop.

---

### Decision · Program_Chairs · 2022-07-06

Accept (Poster)